



# Estimation of biogenic volatile organic compound (BVOC) emissions in China using WRF–CLM–MEGAN coupled model

Lifei Yin[1], Zhenying Xu[1], Mingxu Liu[1], Tingting Xu[1,2], Tiantian Wang[1], Wenling Liao[1], Mengmeng Li[3], Xuhui Cai[1], Ling Kang[1], Hongsheng Zhang[4], Yu Song[1]

[1]State Key Joint Laboratory of Environmental Simulation and Pollution Control, Department of Environmental Science, Peking University, Beijing 100871, China
[2]Environmental College, Chengdu University of Technology, Chengdu 610059, China
[3]School of Atmospheric Sciences, Nanjing University, Nanjing 210000, China
[4]Laboratory for Climate and Ocean-Atmosphere Studies, Department of Atmospheric and Oceanic Sciences, School of Physics,
Peking University, Beijing 100871, China

*Correspondence to*: Yu Song (songyu@pku.edu.cn)

**Abstract.** Biogenic volatile organic compounds (BVOCs) emitted by terrestrial vegetation significantly influence the atmospheric chemistry and global climate. Previous studies calculated BVOC emissions outside the comprehensive terrestrial ecosystem processes and oversimplified the representation of canopy environments which exert substantial impacts on BVOC

emissions. The sophisticated land surface model CLM (Community Land Model) reproduces essential ground and canopy characteristics and can calculate BVOC emissions as a step of integrated biogeochemical processes. In this study, the land surface scheme CLM version4 (CLM4) of WRF (Weather Research and Forecasting model) was used to estimate BVOC emissions in China. Based on highly-resolved meteorological outputs derived from WRF, CLM4 computed real-time physical and biological variables to drive MEGAN (Model of Emissions of Gases and Aerosols from Nature), a BVOC estimation

algorithm embedded within biogeochemistry component of CLM4, to calculate plant emission flux. MODIS (Moderate Resolution Imaging Spectroradiometer) land-use data with high resolutions was introduced in WRF to replace the outdated land surface parameters. An emission inventory of isoprene and monoterpenes (including α-pinene, β-pinene, 3-carene, t-β-ocimene, limonene, sabinene and myrcene) with high spatiotemporal resolution (12 ×12 km, hourly) was established for the year 2018. The annual BVOC emission in China was 14.7 Tg C, in which isoprene contributes about 78.3 % (11.5 Tg C),

followed by α-pinene (1.2 Tg C) and β-pinene (0.7 Tg C). Due to the strong emission capacity and large areas, broadleaf forests contribute to 76.8 % of total isoprene emission and 72.1 % of monoterpenes emission, respectively. BVOC emissions showed marked seasonal and diurnal patterns with the peak emission occurring in summer and midday. Spatially, high emissions of BVOC were mainly concentrated in southern and northeastern China, as well as the Qinling Mountains in central China, accounting for 91.4 % of national emission. Guangxi, Yunnan and Hunan provinces are significant emitters due to large area

of vegetation with high emission rate and favored environmental conditions. The emission estimates are compared to past modeling results, field measurements and further evaluated against top-down isoprene emission estimates. Generally, the coupled mode produced a reasonable simulation in both emission amounts and the spatiotemporal distribution of BVOCs. The



WRF–CLM–MEGAN coupling framework could be further integrated with atmospheric chemistry model to investigate BVOC chemistry and their effects on regional pollution and climate.

## 1. Introduction

Globally speaking, biogenic volatile organic compounds (BVOCs) emitted by terrestrial vegetation are 1150 Tg C yr$^{-1}$, corresponding to about 90 % of the emission total (Guenther et al., 1995). Regionally, although anthropogenic sources are important in urban areas, BVOCs emissions contribute significantly to the total VOC emissions of both urban and rural areas (Benjamin et al., 1997). Many BVOC species are actively involved in the atmospheric chemistry and have substantial impact on tropospheric oxidation, aerosol concentration and the global carbon cycle (Fehsenfeld et al., 1992; Andreae and Crutzen, 1997). BVOCs are therefore a crucial component of the earth system and quantitative estimates of their emissions are required for further exploring their impacts on climate.

It has been demonstrated that a number of ambient and canopy factors modify the capacity of plant to emit BVOC species, such as leaf temperature, leaf phenology, radiation, soil moisture and atmospheric composition (Guenther et al., 2006). However, most studies oversimplified the representation of canopy environments and calculated BVOC emissions as an isolated step outside the whole terrestrial ecosystem processes (Guenther et al., 1995; Guenther et al., 2006; Sindelarova et al., 2014; Müller et al., 2008; Sakulyanontvittaya et al., 2008). For example, BVOC emissions are highly dependent on leaf temperature, while some studies used ambient temperature instead or calculated leaf temperature using independent canopy environment models. Although leaf temperature is primarily dependent on ambient temperature, it is also influenced by other ecological conditions that cannot be reproduced well in canopy environment models, such as humidity, evapotranspiration and soil moisture (Guenther et al., 2006). Additionally, using the dominant vegetation type in a grid for simulation, many studies failed to consider the ecological differences among vegetation types within a grid cell.

Only a few studies coupled BVOC emissions within land surface process. Levis et al. (2003) incorporated BVOC algorithms within Community Land Model version 2 (CLM2) as one step toward integrating biogeochemical processes in the land surface parameterizations. CLM2 formalizes and quantifies concepts of ecological climatology with representations of essential land and vegetation processes, which is better to drive BVOC algorithms. However, CLM2 is originally coupled within earth climate models, the resolution of which ranges over hundreds of kilometers and is too coarse to simulate regional weather and climate accurately (Bonan et al., 2002; Jin et al., 2010).

CLM4 (denoted hereafter as CLM) was coupled and released with the Weather Research and Forecasting model (WRF), a mesoscale numerical model designed to simulate or predict regional weather and climate, since version 3.5 as one of the land surface scheme options. The coupled model made it possible to estimate regional BVOC emissions within a comprehensive ecological climatology framework. CLM consists of components related to land biogeophysics, hydrologic cycle, biogeochemistry, human dimensions and ecosystem dynamics. A wide range of ecological and physiological processes which influence BVOC emissions are simulated in different component, providing real-time variables to drive BVOC algorithms.





For instance, leaf temperature and canopy solar radiation are calculated at every time step in the biogeophysics module. The hydrologic cycle includes representations of soil moisture, transpiration and canopy evaporation, which are directly linked to the biogeophysics and also affect leaf temperature simulations. Moreover, as a sophisticated ecological framework, CLM also characterizes other land surface processes that influence BVOC emissions but have not been parameterized in current BVOC algorithms, such as photosynthesis and stomatal conductance (Oleson et al., 2010; Guenther et al., 2000). Through the

comprehensive parameterization of canopy and soil processes, CLM can provide more reasonable canopy environment variables for regional BVOC estimates. However, the coupled mode has seldom been used to estimate regional BVOC emissions.

   Large areas of China are covered by various vegetation. Due to the effects of afforestation and forest management, observed leaf area index (LAI) in eastern China experienced persistent increase over the past three decades (Zhu et al., 2016), enhancing

BVOC emissions. In recent decades, studies have been conducted in China to estimate national or regional BVOC emissions which reported a wide emission range of 4.1–23.4 Tg C yr$^{-1}$ for isoprene and 1.8–5.6 Tg C yr$^{-1}$ for monoterpenes on the national level (Guenther et al., 1995; Klinger et al., 2002; Guenther et al., 2006; Tie et al., 2006; Fu and Liao, 2012; Li et al., 2013; Stavrakou et al., 2014). However, none of these studies estimated emissions within a comprehensive land surface model, leading to large inaccuracies in calculation process and estimations.

In this study, we aimed to use a more reasonable way to estimate BVOC emissions in China. The WRF–CLM coupling framework was used to drive the Model of Emissions of Gases and Aerosols from Nature (MEGAN), the BVOC emission model embedded within CLM land surface scheme. The BVOC emissions were calculated within land surface process using real-time variables provided by CLM. CLM allowed for multiple plant types within a grid cell, thereby accounting for sub-grid ecological differences among vegetation types. Moreover, the atmospheric chemistry and transport module could be

further coupled with the present WRF–CLM framework to achieve a synchronous coupling of emissions, transport and chemistry processes. The dynamic vegetation scheme in CLM could allow WRF to predict future variations in regional vegetation cover and BVOC emissions (Levis et al., 2003; Lawrence et al., 2011). Because we aimed to evaluate the ability of the coupling framework to calculate regional BVOC emissions and develop an inventory for China, the chemistry and dynamic vegetation module was not considered in this study. Highly-resolved Moderate Resolution Imaging Spectroradiometer

(MODIS) land cover and LAI datasets were introduced into WRF to replace outdated land surface parameters. Two primary classes of BVOCs, isoprene ($C_5H_8$) and monoterpenes ($C_{10}H_{16}$) (including α-pinene, β-pinene, 3-carene, t-β-ocimene, limonene, sabinene and myrcene), were considered in this study.

## 2. Methods and data

### 2.1 WRF–CLM–MEGAN coupled model

The CLM was coupled and released with WRF since version 3.5 as one of the land surface scheme options. CLM includes 5 layers for snow, 15 layers for soil and 1 layer for vegetation and has been shown to be accurate in describing soil and vegetation





processes (Bonan et al., 2002; Zeng et al., 2002; Lawrence et al., 2011). In the coupling framework, the BVOC emissions were calculated as part of biogeochemistry within terrestrial ecosystem parameterization. Using highly-resolved meteorological fields provided by WRF, CLM represented comprehensive ecological and physiological processes and computed real-time

environmental factors to drive MEGAN to calculate real-time emission response. Detailed information and algorithms of CLM can be found in Oleson et al. (2010).

MEGAN is a modeling framework for estimation fluxes of biogenic compounds from terrestrial ecosystems using mechanistic algorithms to account for the major known process controlling biogenic emissions. MEGAN estimates emissions ($F_i$, µg C m$^{-2}$ ground area h$^{-1}$) of BVOC species $i$ according to

$$F_i = \gamma_i \times \sum \varepsilon_{i,j} \times \chi_j \qquad (1)$$

Where $\varepsilon_{i,j}$ (µg C m$^{-2}$ h$^{-1}$) is the emission factor (EF) at standard conditions for PFT $j$ with fraction coverage $\chi_{i,j}$. PFT-specific EFs of individual species were determined based on previous studies or literature (Klinger et al., 2002; Bai et al., 2006; Tie et al., 2006; Guenther et al., 2006; Wang et al., 2011; Fu and Liao, 2012; Guenther et al., 2012). Table 1 summarizes the EFs for each vegetation type used in this study.

The emission activity factor for each compound ($\gamma_i$) accounts for emission response to light, leaf temperature, leaf age, soil moisture, LAI and $CO_2$ inhibition. Therein, the canopy solar radiation and leaf temperature was calculated in the biogeophysical component of CLM, and the soil moisture was represented in hydrologic cycle module. The interactions between each component were also taken into account. LAI could be calculated by vegetation dynamics in CLM, which is beyond the scope of this study; therefore, we used satellite-observed LAI data for calculation. In this study, the influence of

$CO_2$ concentration was neglected, so that $\gamma_C$ for all compounds was set to 1. The details of the algorithms could be found in Guenther et al. (2012).

In this study, the simulations with the WRF version 4 were performed on a domain at 12 km horizontal resolution covering China and its surrounding areas with $420 \times 380$ cells in the horizontal direction and 35 layers in the vertical direction, extending from the surface to 50 hPa. The initial meteorological fields and boundary conditions were from the 6 h NCEP (National

Centers for Environmental Prediction) global final analysis with a $1° \times 1°$ spatial resolution. The simulation was conducted for the year 2018, and each run covered 31 days in order to account for the long-term influence of meteorological conditions (temperature and solar radiation) on plant emission ability.

## 2.2 Improvement of land surface parameters in WRF and CLM

The standard WRF initial static field uses the 24-category United States Geological Survey (USGS) global 1 km land cover

map derived from the monthly Advanced Very High Resolution Radiometer (AVHRR) observation, which is based on 1-year data from April 1992 to March 1993. This data could not represent the real surface features due to the rapid land cover changes. In this study, MODIS datasets of land cover (MCD12Q1) for the year 2016 and water mask (MOD44W) for the year 2015, both with a resolution of 500 m, were used to obtain a new land use map by mapping the 17 MODIS land-use categories





defined by the International Geosphere Biosphere Program (IGBP) onto the outdated 24 USGS categories. The coverage

fraction of each land use type in each grid cell was also calculated based on the new land use map. The significantly improved

model performance resulted by replacing land cover data has been revealed in Li et al. (2014b) and Li et al. (2015).

Default CLM prescribes the PFTs composition and abundance of each land cover type of USGS, leading to a geographical-invariant plant distribution that may not match the actual situation (Oleson et al., 2010). We modified the default setting to represent the sub-grid surface heterogeneity with the composition and fraction of 500 m-resolved land cover data within the

12 km-resolved model grid. The 12 vegetation categories in IGBP were converted to 7 PFTs in CLM scheme. Table 2 illustrates the conversion of MODIS land cover into USGS and PFTs. The mixed forests and woody savannas in MODIS data were considered as broadleaf deciduous trees and the savannas in MODIS were identified as shrubs. The conversion was determined according to vegetation distributions presented in three land cover datasets, including GlobeLand30 (Chen et al., 2017), 30m-resolution Forest map for China (Li et al., 2014a) and the Vegetation Atlas of China (Li et al., 2013). The spatial distribution

of seven major PFTs was shown in Fig. 1 (the small island in the South China Sea are not included).

LAI data used in default CLM scheme are updated daily by linearly interpolating between prescribed monthly values. In this study, the MODIS LAI data derived from MCD15A2H version 6 with a spatial resolution of 500 m and temporal resolution of 8 days was introduced in CLM to replace default values and drive MEGAN to estimate the influence of LAI and leaf age on BVOC emission capacities. Instead of using a grid-averaged LAI value for all vegetation types, the PFT-specific LAI was

averaged over the fraction of land area covered by the particular PFT within the grid cell.

In the origin CLM land surface scheme, the calculation of average temperature and radiation of the past days, which is required by MEGAN algorithm, exhibited errors and was commented out. We modified this part of the code in order to consider the long-term influence of meteorological conditions on emission abilities.

## 3 Results and discussions

### 3.1 Evaluation of WRF output

The observed surface temperature of 362 sites in China provided by the National Climatic Data Center (NCDC), available with a temporal resolution of an hour, and observed daily total radiation data of 100 sites in China downloaded from the National Meteorological Information Center (https://data.cma.cn/) were used to evaluate the simulation performance of the temperature at 2 m height (T2) and the downward shortwave radiation (SWDOWN) of WRF. The statistical analyses for four

seasons are displayed in Table 3. The mean error (ME), mean bias (MB), correlation coefficient ($r$) and root-mean-square error (RMSE) of station-averaged hourly T2 series are 1.70, −0.52, 0.98, 2.51°C, respectively. The $r$ value in summer (0.90) is relatively lower than those in spring (0.95), autumn (0.96), winter (0.97), and the simulation shows slight cooling bias in spring, autumn and winter. The ME, MB, $r$, RMSE values of the whole-year SWDOWN series are 71.01, 69.82, 0.86 and 71.11 W m$^{-2}$, and the simulation shows overestimation of 85.37, 86.05, 55.55 and 52.32 W m$^{-2}$ in spring, summer, autumn and winter,

respectively.



The seasonal variation and spatial distributions of simulated seasonally average T2 and SWDOWN are presented in Fig. 2 (excluding small islands in the South China Sea). In general, the simulations of WRF–CLM reproduce the temporal and spatial patterns of T2 and SWDOWN. The highest T2 and SWDOWN values occur in summer, followed by spring and autumn, and then winter. The temperature generally decreases from the south to the north, consistent with terrain height and latitude. The

spatial characteristics of solar radiation are influenced by latitude, altitude and climate change. Overall, the solar radiation reaches a high value near the Tibetan Plateau, and low values occur around the Sichuan Basin. The WRF model can reasonably simulate the spatial pattern of the solar radiation, whereas the amounts are higher than observations, as illustrated in Table 3. The overestimation could be attributed to ignoring radiation effect of aerosols in the model (Markovic et al., 2008; Wild, 2008). Besides, the excess downward solar radiation is a discrepancy that has been found in many other studies, suggesting that the

WRF model has some deficiencies independent of the land surface model that lead to the systematic overestimation (Lu and Kueppers, 2012). Generally, Table 3 as well as Fig. 2 indicate relatively good simulation among most of sites; therefore, the meteorological conditions simulated by WRF–CLM are desirable for driving MEGAN.

## 3.2 BVOC emission budgets

Hourly emissions of 8 chemical species emitted from biogenic sources including isoprene, α-pinene, β-pinene, 3-carene, t-β-

ocimene, limonene, sabinene and myrcene were calculated by WRF-CLM-MEGAN model on a 12 km × 12 km grid for the year 2018. In the following section, all the results are measured as carbon weights of the constituent compounds, unless stated otherwise.

The China land area can be subdivided into six regions as shown in Fig. S1 (the small island in the South China Sea are not included). Regional and national annual totals of individual species are listed in Table 4. The annual amount emitted for all

listed BVOCs reaches 14.7 Tg C yr$^{-1}$ with isoprene accounting for 78.3 % (11.5 Tg) and sum of monoterpenes for 21.7 % (3.2 Tg). Regarding the relative contributions of species considered in the sum of monoterpenes, α-pinene (1.2 Tg) is the largest contributor, accounting for about 8.4 % of the total BVOC emissions and 38.8 % of the total monoterpenes emissions. The other main contributors are β-pinene (0.7 Tg) and limonene (0.4 Tg), which are responsible for 21.6 % and 13.3 % of the annual monoterpenes emission, respectively. The emissions of other monoterpene species are negligible: t-β-ocimene (0.3 Tg),

3-carene (0.2 Tg), sabinene (0.2 Tg), myrcene (0.1 Tg).

The emission ability of BVOC varies substantially between and within PFTs (Guenther et al., 2006). The field measurements indicated that broadleaf forests (especially *Quercus*, *Populus* and *Eucalyptus*) and shrubs have high isoprene emission capacities, while conifer species (especially *Pinus* and *Picea*) favor terpene emission over isoprene. Crops and grass are considered to have low BVOC emitting capacities (Guenther, 2013; Bai et al., 2016; Klinger et al., 2002; Bai et al., 2015). The

relative contribution of each considered PFT to the national BVOC totals is shown in Table 5.

For both isoprene and monoterpenes, the predominant sources of emissions are the broadleaf deciduous forests with a relative contribution of 64.5 % and 60.6 %, respectively. Shrubs rank second of the emission contribution, accounting for 19.9 % of total BVOC emissions, followed by broadleaf evergreen trees (12.1 %). In sum, broadleaf trees are responsible for 76.8 % of





isoprene emission and 72.1 % of monoterpenes emission, respectively. The contribution of needleleaf forests to monoterpenes

emission (6.4 %) is higher than that to isoprene (0.5 %), in agreement with the in situ experiment results. Grass and Crop are minor BVOC sources with only 1.3 % of total isoprene emission and 1.1 % of monoterpene emission, respectively.

## 3.3 Spatial distribution of BVOC emission budget

As described in section 2.1, the spatial variation of BVOC emissions is determined by the distribution of tree species and meteorological conditions.

The spatial distributions of annual emissions of isoprene and monoterpenes are displayed in Fig.3 (excluding small islands in the South China Sea). As shown in Table 4 and Fig.3, BVOCs are mainly emitted in southwestern China, which contributes to about 35.4 % to the national total, followed by central China (25.6 %) and southeastern and northeastern regions contributing with 20.1 % and 10.4 %, respectively. Less than 10 % of national BVOCs are emitted in the northern and northwestern China. The southern and northeastern regions of China, as well as the Qinling Mountains in central China exhibit high emission

budgets, accounting for 91.3 % of the national isoprene emission and 91.8 % of the monoterpenes emission. The top-ranking isoprene contributor is Guangxi, with approximately 1.3 Tg C emitted yearly, followed by Yunnan and Hunan (both 1.1 Tg). According to Fig.1 and the survey results from the Plant Research Institute, these areas are covered by vegetation species with high emission capacity of isoprene (broadleaf forests, shrub) or monoterpenes (coniferous forest). Northeast China is primarily covered by deciduous coniferous forests (mainly *Larix gmelini*) and deciduous broadleaf forests (mainly *Quercus mongolica,*

*Tilia Mongolia* and *Betula platyphylla*). Large areas of evergreen coniferous forests (mainly *Pinus massoniana* and *Cunninghamia lanceolata*) and shrubs are found in Southeast China, and the main plant genera in Southwest China are evergreen tree species, including evergreen broadleaf forests (e.g. *Quercus aquifolioides*), evergreen coniferous forests (*Picea likiangensis var.b*alfouriana and *Pinus yunnanensis*) and shrubs. The Qinling Mountains are covered by large areas of deciduous broadleaf forests (mainly *Quercus variabilis* and *Quercus liaotungensis*).

However, the results also illustrate a contradiction between the high plant coverage and the relatively low isoprene emission in Sichuan Province. With abundant broadleaf deciduous tree and shrub, Sichuan emits 0.7 Tg C annually, which could be attributed to the low temperatures in the west of Sichuan and low solar radiation in the east of Sichuan (Fig. 2). Similarly, the northeastern China, covered by large areas of broadleaf deciduous forests, shows lower emission amounts (9.2 % of isoprene and 14.5 % of monoterpenes) when compared with the Qinling mountains (26.4 % of isoprene and 22.5 % of monoterpenes).

This results could be partly explained by the higher latitude that leads to lower temperature and light intensity in the northeastern China (as shown in Fig. 2) and thus reduces the vegetation ability of producing and emitting BVOC.

The regions covered by large area of crop or grassland, such as the North China Plain and Inner Mongolia, release extremely small amount of BVOC due to the low BVOC emitting capacities of crop and grass.





### 3.4 Temporal variations in BVOC emissions

BVOC emissions exhibit strong seasonal variation as a consequence of the substantial differences in monthly environmental conditions, such as temperature, solar radiation and plant water stress. The average seasonal emission fluxes of isoprene and monoterpenes are presented in Figs. 4 and 5 (the small island in the South China Sea are not included), and the temporal profile of national monthly emissions of individual species is presented in Fig. 6. Because of the highest light intensities, temperatures and plant biomass density, the emission rates peak in summer (from June to August) and around 55.1 % (8.1 Tg C) of the total

annual BVOC emissions are released during this period. In spring (from March to May), deciduous trees started to develop leaves and the increasing foliage density enhanced the BVOC emission flux. Moreover, less precipitation in spring resulted in inadequate soil moisture, which could decrease BVOC emission rates (Pegoraro et al., 2004). From September, emissions decrease rapidly because of the loss of biomass density and decreasing ambient temperature and solar radiation intensity. The total emission in autumn (from September to November, 3.5 Tg) is slightly higher than that in Spring (2.7 Tg), which could be

explained by the higher leaf area (as shown in Fig.2) and a corresponding higher emission capacity in autumn. In addition, taking the long-term influence of temperature and solar radiation into account also contributes to a higher emission in autumn while a lower value in spring. In winter, ambient temperature and radiation intensity reaches the lowest value for the year and results in the lowest emission rates. Besides, as more than 90 % of BVOC emissions are produced and emitted from plant foliage, extremely low foliage density in winter leads to negligible BVOC emissions (0.4 Tg).

Aside from being seasonal, the natural emissions exhibit an obvious diurnal cyclical pattern. The diurnal variations of temperature, solar radiation and emission of isoprene and monoterpenes in summer are shown in Fig. 7. Simulated emissions of isoprene and monoterpene rise rapidly in the morning and reach the peak in the afternoon (13:00~14:00, Beijing Time), as a consequence of the increasing plant metabolism at times with relatively high light intensity and temperature, which lead to higher activity of BVOC production. Emissions of monoterpenes are temperature dependent but not highly dependent on light

while isoprene emissions are strong light-dependent. As a result, monoterpene emissions maintain a relatively high level during the night while isoprene emissions cease at nighttime. However, since specific light-dependent factors for various compounds, including monoterpenes, are used in the model, the magnitude of the diurnal variation of monoterpene emissions is larger than that estimated by past studies that considered monoterpene emissions as solely temperature dependent (Tie et al., 2006).

### 3.5 Comparisons with field measurements and previous budgets estimations

There are little intensive measurements of BVOC fluxes in China due to the difficulties in techniques of large-scale and long-term measurements. Hence, we just evaluate the inventory against limited results of canopy-scale measurements conducted at different sites in China.

Using eddy covariance technique, Baker et al. (2005) measured isoprene fluxes in Xishuangbanna, Yunnan Province (21.92°N, 101.27°E) and the daytime-averaged fluxes were found to be 1 mg C $m^{-2}$ $h^{-1}$ during the wet season (July 2002). Model results

show that the daytime averaged isoprene flux in July was 2.1 mg C $m^{-2}$ $h^{-1}$ at this site, higher than the measured results by a





factor of 2. A field measurement was conducted at Dinghushan (23.17°N, 112.53°E), Guangdong Province using the Relaxed Eddy Accumulation (REA) technique to quantify BVOC emission flux from a subtropical evergreen mixed forest in fall 2010 (Situ et al., 2013). As a result, the isoprene emission fluxes ranged between 0.002 and 0.215 mg m$^{-2}$ h$^{-1}$ and monoterpenes fluxes ranged between 0.063 and 0.313 mg m$^{-2}$ h$^{-1}$. Model simulations show that the average emission fluxes of daytime in

autumn were 0.351 mg m$^{-2}$ h$^{-1}$ for isoprene and 0.052 mg m$^{-2}$ h$^{-1}$ for monoterpenes, which are close to the observed data range.

Based on REA technique, emissions of isoprene and monoterpenes of a temperate forest in Changbai Mountain (42.4°N, 128.1°E) were measured during the summer seasons (June, July, August, September) in 2010 and 2011 (Bai et al., 2015). The mean isoprene emission flux was 0.889 mg m$^{-2}$ h$^{-1}$ and the mean total monoterpene emission flux was 0.143 mg m$^{-2}$ h$^{-1}$, and

our results estimated emission fluxes of 2.032 for isoprene and 0.453 for monoterpene. Using the same method, Bai et al. (2016) measured emissions from a bamboo (*Phyllostachys violascenes*) plantation in Zhejiang Province (30.3°N, 119.57°E) and the isoprene emission flux averaged in July and August was 1.91 mg m$^{-2}$ h$^{-1}$, which is close to our estimated value of 2.03 mg m$^{-2}$ h$^{-1}$.

Generally, the agreement of our simulated emissions with most observations is satisfactory, because the meteorological

conditions at the specific times and sites can significantly differ from that of the simulation period, and some special conditions (such as severe drought) at the site may not be reproduced well by the model (Stavrakou et al., 2014). In addition, the measurement technique may underestimate the BVOC flux. For example, Bowling et al. (1999) suggested that mixing of samples would lead to an underestimated flux. Overall, the field measurements of BVOC flux can be very different from the model estimate, and more field experiments are needed to accurately quantify the BVOC emission fluxes and to further

evaluate the model performance.

Because of the differences in algorithms and input data applied, our results may differ greatly from previous budget estimations. As illustrated in Table 6, the annual emission budgets estimated by this study fall in the range of past studies and is close to results of Guenther et al. (2006) and Fu and Liao (2012), while lower than the result of Guenther et al. (1995) by 31.3 % and higher than that of Klinger et al. (2002) by 93.4 %, which were all based on the outdated G95 algorithms (Guenther et al.,

1995). This discrepancy between results of Klinger et al. (2002) and this study is mainly due to the differences in vegetation distribution and Klinger et al. (2002) can only reflect the natural emission level of vegetation during the last decade. Li et al. (2012) suggested that the use of outdated USGS land-use distribution in Tie et al. (2006) should be responsible for the underestimation of BVOC emissions. Considering the same land use and LAI datasets used for calculation, the discrepancy between results of Fu and Liao (2012) and this study could be attributed to the differences in algorithms. Similar to most past

studies, Fu and Liao (2012) considered only one dominant land cover type within a grid cell and applied grid-averaged LAI for calculation, ignoring the extreme differences in emission capacities between tree species. The use of average LAI might overestimate emissions from plants with small leaf area (e.g., grass, coniferous trees) and underestimate emissions from plants with large leaf area (broadleaf trees, shrub). The use of WRF–CLM in this study accounted for vegetation diversity within grid can produce a more reasonable emission estimates.





Formaldehyde (HCHO), as a major intermediate product in the degradation of isoprene in the atmosphere, has been widely used as a proxy for estimates isoprene emissions. Fu et al. (2007) used a continuous 6-year record (1996–2001) of Global Ozone Monitoring Experiment (GOME) HCHO columns and made a top-down estimates for isoprene as 12.7 Tg yr$^{-1}$ in China, which is comparable to the our model outputs. However, Stavrakou et al. (2014) inferred isoprene emissions by inversion of GOME-2 HCHO columns and the satellite-derived emissions were found to decrease from 8.6 Tg in 2007 to 6.5 Tg in 2012,

lower than emissions in this study by 33.7 %–76.9 %. Aside from the influence of difference meteorological conditions and land cover changes during the past years, the reliability of satellite-based constraints also needs to be improved because that the HCHO is affected by non-isoprene sources plus uncertainty and spatial smearing in isoprene-formaldehyde relationship (Fu et al., 2019).

## 4 Uncertainty

Uncertainties in both model inputs and modeling framework could affect the accuracy of estimation. There are three main inputs to the MEGAN: land cover data (vegetation distribution and LAI), emission factors and meteorological data.

The basal emission factors have been identified as the most important uncertainty source (Situ et al., 2014; Wang et al., 2016; Zheng et al., 2010). We used the localized emission factors based on BVOC emission rate measurements from previous studies. However, local studies measuring the basal emission rate of BVOCs cover only a small fraction of tree species. Further

measurements for emission rates of plantations in China are required. In addition, since the PFT-specific emission rates are assigned to each grid according to the land cover data, an accurate vegetation distribution map with higher spatial resolution and more detailed classification of tree species is expected to improve the accuracy of emission rate assignment. Wang et al. (2011) investigated the impact of meteorological conditions and indicated that the simulation bias of temperature and solar radiation resulted in 26.7 % and 50.7 % uncertainties to isoprene, respectively. Wang et al. (2018) adopted three LAI datasets

to study the effect of the LAI input on BVOC emissions and indicated that the discrepancies between different LAI inputs do not obviously affect the estimates, similar to the results of Zheng et al. (2010).

The emission algorithm is also an important source of uncertainty. Although the dominant processes controlling BVOC emission have been considered in MEGAN algorithm, substantial uncertainties remain. For example, the influence of concentration of $CO_2$ and ozone has not been quantified in this study. It has been indicated that the empirical coefficients used

for computation contribute to uncertainties in estimates as well (Zheng et al., 2010; Situ et al., 2014).

## 5 Conclusion

This study estimated the emission budgets and spatial-temporal patterns of BVOC in China in 2018 using the WRF-CLM-MEGAN modeling system. CLM characterizes comprehensive land surface processes including biogeophysics, biogeochemistry, hydrologic cycle and ecosystem dynamics, which are strongly linked to the BVOC emissions. Through the



parameterization of these ecological and physiological processes, CLM can provide more reasonable canopy environment variables required by BVOC algorithms. Highly-resolved MODIS land-use data were introduced in WRF to replace the outdated land surface parameters and improve model performance.

Isoprene ($C_5H_8$) and monoterpenes ($C_{10}H_{16}$) (including α-pinene, β-pinene, 3-carene, t-β-ocimene, limonene, sabinene and myrcene) emissions were calculated and analyzed. The total amount of BVOCs emitted in China was estimated to be 14.7 Tg

C yr$^{-1}$, with isoprene and monoterpenes accounting for 78.3 % and 21.7 % of the totals, respectively. Spatially, the hot spots of BVOC emissions includes the southern and northeastern regions of China, as well as Qinling Mountains in central China, owing to the dense plant cover in these regions. Due to the strong emission capacity and large coverage, broadleaf deciduous forest has been identified as dominate emitter of isoprene and monoterpenes, accounting for 64.5 % of total isoprene emissions and 60.6 % of total monoterpenes emissions, respectively. Coniferous trees were responsible 5.6 % monoterpenes, while

emitted little isoprene (0.5 %). As the temperature and solar radiation varied substantially, BVOC emissions correspondingly exhibited significant seasonal variability: the summer time contributed 55.1 % of the total emissions while the winter only contributed 2.9 %. The diurnal variation mainly follows the solar radiation and temperature pattern and daily maximum occurs at about 13:00~14:00 (Beijing Time).

Our results are at the upper end of values presented in previous modeling estimates, which can be partly explained by the

335 incorporation of CLM, which provides more reasonable canopy characterizations. Comparison of earlier studies on national isoprene estimates has shown that isoprene emissions can vary within a factor of 2. The simulated variations in spatial and temporal patterns are in excellent agreement with earlier studies.

Results of this study are compared to flux measurements at four sites located in the southwestern, southeastern and northeastern China. The comparison shows that the model tends to overestimate the isoprene fluxes by 6.2 %~128.6 % and monoterpene

fluxes by a factor of 2. The different meteorological conditions at the specific times and sites that measurements were taken are different from and that of the simulation period can lead to substantial discrepancy between observations and simulations. Moreover, mechanical failures in field measurements may result in underestimation of the BVOC flux. Two top-down results estimated by inverse modeling using formaldehyde as a proxy for VOC emissions are used to evaluate our results and the discrepancy ranges between −10.4 % and 76.9 %. Although using a model to estimate BVOC emissions is a key study approach,

additional ambient observed data are needed to further evaluate the model performance. On the basis of this study, we can further couple this framework with chemistry module to investigate the role of BVOCs in regional and global atmosphere chemistry, carbon cycle and climate change.

*Data availability.* MCD12Q1 product can be freely accessed at https://doi.org/10.5067/MODIS/MCD12Q1.006. MOD44W data can be download from https://doi.org/10.5067/MODIS/MOD44W.006. MCD15A2H data can be found at

350 https://doi.org/10.5067/MODIS/MCD15A2H.006.





*Author contributions.* This work was designed by YS, XC, LK and HZ, and performed by LY, ZX, ML, TX, TW, WL and ML. LY and YS led the writing of the papers and prepared the figures with contributions from all co-authors.

*Competing interests.* The authors declare that they have no conflict of interest.

*Acknowledgements.* The MODIS land cover data (MCD12Q1), water mask data (MOD44W) and leaf are index product
(MCD15A2H) were provided by Land Process Distributed Active Archive Center (LPDAAC), USA. This work was supported
by the National Natural Science Foundation of China (NSFC) (41675142, 91644212).



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



**Table 1 BVOC emission factors (EF, µg C m$^{-2}$ h$^{-1}$) for each plant functional type (PFT).**

| PFT | NE_EV [a] | NE_DE [a] | BR_EV [a] | BR_DE [a] | Shrub | Grass | Crop |
|---|---|---|---|---|---|---|---|
| ISO [b] | 1270.2 | 1100.0 | 3565.4 | 4638.8 | 1280 | 231.9 | 90.0 |
| API [b] | 317.6 | 801.4 | 169.8 | 147.3 | 35.9 | 0.9 | 2.0 |
| BPI [b] | 190.5 | 314.3 | 34.0 | 47.9 | 17.9 | 0.7 | 1.5 |
| 3-CAR [b] | 101.6 | 125.7 | 11.3 | 11.0 | 12.0 | 0.1 | 0.3 |
| OCI [b] | 44.5 | 94.3 | 42.4 | 44.2 | 17.9 | 0.9 | 2.0 |
| LIM [b] | 63.5 | 204.3 | 22.6 | 29.5 | 12.0 | 0.3 | 0.7 |
| SAB [b] | 44.5 | 62.9 | 22.6 | 18.4 | 8.4 | 0.3 | 0.7 |
| MYR [b] | 44.5 | 94.3 | 22.6 | 11.0 | 6.0 | 0.1 | 0.3 |

[a] NE_EV: Needleleaf Evergreen Tree; NE_DE: Needleleaf Deciduous Tree; BR_EV: Broadleaf evergreen Tree; BR_DE: Broadleaf
Deciduous Tree.
[b] ISO: isoprene, API: α-pinene, BPI: β-pinene, 3-CAR: 3-carene, OCI: ocimene, LIM: limonene, SAB: sabinene, MYR: myrcene.



**Table 2 The mapping of land cover types of MODIS data (MOD12Q1) to USGS classifications and plant functional types (PFTs)**
**used in CLM.**

| MODIS | | USGS | | CLM |
|---|---|---|---|---|
| Category | Land-use Description | Category | Land-use Description | Plant Functional Type (PFT) |
| 0 | Water | 16 | Water bodies | - |
| 1 | Evergreen Needleleaf Forest | 14 | Evergreen Needleleaf forest | Needleleaf Evergreen Tree |
| 2 | Evergreen Broadleaf Forest | 13 | Evergreen Broadleaf forest | Broadleaf Evergreen Tree |
| 3 | Deciduous Needleleaf Forest | 12 | Deciduous Needleleaf forest | Needleleaf Deciduous Tree |
| 4 | Deciduous Broadleaf Forest | 11 | Deciduous Broadleaf forest | Broadleaf Deciduous Tree |
| 5 | Mixed Forest | 15 | Mixed Forest | Broadleaf Deciduous Tree |
| 6 | Closed Shrubland | 8 | Shrubland | Shrub |
| 7 | Open Shrubland | 9 | Mixed Shrubland/Grass | Shrub |
| 8 | Woody savanna | 11 | Deciduous Broadleaf forest | Broadleaf Deciduous Tree |
| 9 | Savanna | 10 | Savanna | Shrub |
| 10 | Grassland | 7 | Grassland | Grass |
| 11 | Permanent wetland | 17 | Herb. Wetland | - |
| 12 | Cropland | 5 | Crop/Grass Mosaic | Crop |
| 13 | Urban and Built-up | 1 | Urban | - |
| 14 | Cropland/Natural Vegetation Mosaic | 6 | Crop/Wood Mosaic | Crop |
| 15 | Snow and Ice | 24 | Snow or Ice | - |
| 16 | Barren or Sparsely Vegetated | 19 | Barren or Sparsely Vegetated | - |
| 254 | Unclassified | 25 | No Data | - |





**Table 3 Verification statistics of temperature at 2 m height (T2) and downward shortwave radiation (SWDOWN).**

| Variable | Season | Mean | | ME [b] | MB [b] | r [b] | RMSE [b] |
|---|---|---|---|---|---|---|---|
| | | Obs. [a] | Sim. [a] | | | | |
| T2 (°C) | Spring | 13.99 | 13.95 | 1.54 | −0.04 | 0.95 | 2.26 |
| | Summer | 23.60 | 23.65 | 1.64 | 0.05 | 0.60 | 2.54 |
| | Autumn | 12.70 | 12.10 | 1.47 | −0.60 | 0.96 | 2.16 |
| | Winter | −1.34 | −2.84 | 2.17 | −1.50 | 0.97 | 3.00 |
| | Year | 12.23 | 11.70 | 1.70 | −0.52 | 0.98 | 2.51 |
| SWDOWN (W m$^{-2}$) | Spring | 176.39 | 261.75 | 85.98 | 85.37 | 0.64 | 90.40 |
| | Summer | 192.25 | 278.29 | 86.50 | 86.05 | 0.59 | 92.00 |
| | Autumn | 129.03 | 184.58 | 57.86 | 55.55 | 0.59 | 61.89 |
| | Winter | 92.04 | 144.36 | 53.69 | 52.32 | 0.71 | 57.60 |
| | Year | 147.43 | 217.25 | 71.01 | 69.82 | 0.86 | 77.11 |

[a] Obs.: Mean observed value, Sim.: Mean simulated value;
[b] ME: Mean Error, MB: Mean Bias, r: correlation coefficient, RMSE: Root Mean Square Error.



**Table 4 Province-specific estimated BVOC emission budgets (Gg C).**

| Region/Province | ISO [a] | MT [b] | | | | | | | | | T_Pro [c] |
|---|---|---|---|---|---|---|---|---|---|---|---|
| | | API [a] | BPI [a] | 3-CAR[a] | OCI [a] | LIM [a] | SAB [a] | MYR [a] | T_MT [c] | |
| **Northwest** | **581.4** | **55.4** | **30.6** | **8.3** | **13.7** | **18.6** | **7.6** | **4.5** | **138.7** | **720.1** |
| Xinjiang | 33.2 | 1.3 | 1.1 | 0.5 | 0.7 | 0.6 | 0.3 | 0.2 | 4.7 | 37.9 |
| Gansu | 93.1 | 8.5 | 5.0 | 1.6 | 2.1 | 3.1 | 1.2 | 0.8 | 22.3 | 115.4 |
| Ningxia | 3.4 | 0.1 | 0.1 | 0.0 | 0.1 | 0.0 | 0.0 | 0.0 | 0.4 | 3.8 |
| Qinghai | 9.7 | 0.3 | 0.3 | 0.1 | 0.2 | 0.1 | 0.1 | 0.0 | 1.0 | 10.7 |
| Shaanxi | 441.9 | 45.2 | 24.0 | 6.2 | 10.7 | 14.8 | 5.9 | 3.6 | 110.3 | 552.2 |
| **Northeast** | **1062.4** | **181.9** | **105.7** | **32.2** | **37.9** | **65.8** | **22.9** | **16.3** | **462.6** | **1525.0** |
| Inner Mongolia | 229.6 | 65.7 | 38.9 | 13.5 | 12.0 | 24.5 | 7.7 | 6.6 | 168.8 | 398.4 |
| Heilongjiang | 438.3 | 75.8 | 43.6 | 12.5 | 16.5 | 27.1 | 9.8 | 6.4 | 191.7 | 630.0 |
| Jilin | 247.8 | 26.6 | 15.0 | 3.6 | 5.9 | 9.2 | 3.4 | 2.0 | 65.9 | 313.7 |
| Liaoning | 146.7 | 13.8 | 8.2 | 2.5 | 3.4 | 5.0 | 2.0 | 1.2 | 36.2 | 182.9 |
| **North** | **418.3** | **44.1** | **26.3** | **10.7** | **13.9** | **16.1** | **7.5** | **4.7** | **123.1** | **541.4** |
| Beijing | 32.8 | 2.8 | 1.6 | 0.7 | 0.8 | 1.0 | 0.5 | 0.3 | 7.8 | 40.6 |
| Shanxi | 92.9 | 11.4 | 6.6 | 2.6 | 3.3 | 4.1 | 1.8 | 1.2 | 31.0 | 124.0 |
| Hebei | 132.2 | 10.6 | 6.8 | 3.2 | 3.6 | 4.2 | 2.0 | 1.3 | 31.7 | 163.9 |
| Shandong | 14.9 | 1.2 | 1.1 | 0.6 | 0.8 | 0.5 | 0.3 | 0.2 | 4.6 | 19.5 |
| Tianjin | 1.0 | 0.1 | 0.1 | 0.0 | 0.0 | 0.0 | 0.0 | 0.0 | 0.3 | 1.3 |
| Henan | 144.5 | 18.0 | 10.1 | 3.6 | 5.3 | 6.1 | 2.8 | 1.7 | 47.6 | 192.1 |
| **Central** | **3048.2** | **282.9** | **153.4** | **52.3** | **69.8** | **93.4** | **40.5** | **27.2** | **719.5** | **3767.7** |
| Hubei | 698.3 | 65.8 | 35.1 | 10.7 | 16.7 | 21.8 | 9.2 | 5.7 | 164.9 | 863.2 |
| Anhui | 305.8 | 27.0 | 15.4 | 5.2 | 6.8 | 9.0 | 3.9 | 2.5 | 69.7 | 375.5 |
| Hunan | 1100.0 | 101.9 | 55.9 | 18.8 | 25.2 | 34.1 | 14.5 | 9.6 | 260.1 | 1360.1 |
| Jiangxi | 944.1 | 88.2 | 47.1 | 17.6 | 21.1 | 28.5 | 12.9 | 9.5 | 224.8 | 1168.9 |
| **Southwest** | **4063.6** | **433.8** | **249.0** | **93.5** | **104.0** | **151.7** | **64.4** | **45.5** | **1141.9** | **5205.5** |
| Tibet | 134.5 | 20.5 | 12.1 | 4.7 | 3.5 | 6.4 | 2.8 | 2.5 | 52.45 | 187.0 |
| Sichuan | 691.0 | 69.1 | 43.0 | 15.9 | 17.0 | 25.7 | 10.3 | 6.8 | 187.6 | 878.6 |
| Chongqing | 261.0 | 25.8 | 14.8 | 5.0 | 6.5 | 9.1 | 3.7 | 2.4 | 67.4 | 328.4 |
| Yunnan | 1092.0 | 133.3 | 77.6 | 29.8 | 30.3 | 46.0 | 19.6 | 14.4 | 351.0 | 1443.0 |
| Guizhou | 536.1 | 55.0 | 32.2 | 11.8 | 14.2 | 20.3 | 8.3 | 5.4 | 147.2 | 683.3 |



**Table 4 Continued.**

| Region/Province | ISO [a] | MT [b] | | | | | | | | | T_Pro [c] |
| | | API [a] | BPI [a] | 3-CAR[a] | OCI [a] | LIM [a] | SAB [a] | MYR [a] | T_MT [c] | |
|---|---|---|---|---|---|---|---|---|---|---|
| Guangxi | 1349.0 | 130.1 | 69.3 | 26.3 | 32.6 | 44.2 | 19.7 | 14.1 | 336.3 | 1685.3 |
| **Southeast** | **2353.9** | **241.5** | **126.1** | **45.3** | **55.5** | **78.1** | **34.9** | **26.1** | **607.4** | **2961.3** |
| Jiangsu | 39.2 | 2.7 | 1.9 | 1.2 | 1.2 | 1.2 | 0.6 | 0.4 | 9.2 | 48.4 |
| Shanghai | 1.0 | 0.1 | 0.0 | 0.0 | 0.0 | 0.0 | 0.0 | 0.0 | 0.2 | 1.2 |
| Zhejiang | 402.9 | 44.6 | 26.3 | 8.9 | 9.5 | 14.8 | 6.0 | 4.2 | 114.3 | 517.2 |
| Fujian | 629.0 | 65.7 | 32.6 | 9.9 | 14.0 | 20.4 | 8.9 | 6.7 | 158.1 | 787.1 |
| Guangdong | 879.6 | 87.2 | 45.3 | 18.0 | 21.4 | 29.1 | 13.4 | 10.1 | 224.4 | 1104.0 |
| Macao | 1.4 | 0.2 | 0.1 | 0.0 | 0.0 | 0.0 | 0.0 | 0.0 | 0.4 | 1.8 |
| Hong Kong | 1.1 | 0.1 | 0.1 | 0.0 | 0.0 | 0.0 | 0.0 | 0.0 | 0.3 | 1.4 |
| Hainan | 285.8 | 27.0 | 13.5 | 5.0 | 6.8 | 8.6 | 4.0 | 2.9 | 67.7 | 353.5 |
| Taiwan | 113.9 | 13.9 | 6.4 | 2.3 | 2.5 | 3.9 | 1.9 | 1.8 | 32.8 | 146.7 |
| **Nation** | 11528 | 1240 | 691 | 242 | 295 | 424 | 178 | 124 | 3193 | 14721 |

[a] Refer to Table 1;
[b] MT: Monoterpenes;
[c] T_MT: Regional or provincial monoterpene totals, T_Pro: Regional or provincial BVOC totals, including isoprene and monoterpenes.





**Table 5 The BVOC emission budgets of each plant functional type (PFTs) (Gg C).**

| PFTs | ISO [a] | MT [a] | | | | | | | | T_ALL [b] |
| | | API [a] | BPI [a] | 3-CAR [a] | OCI [a] | LIM [a] | SAB [a] | MYR [a] | T_MT [b] | |
| --- | --- | --- | --- | --- | --- | --- | --- | --- | --- | --- |
| NE_EV [a] | 59.3 | 35.6 | 38.6 | 20.6 | 3.4 | 12.9 | 5.0 | 5.0 | 121.0 | 180.3 |
| NE_DE [a] | 4.2 | 33.7 | 20.0 | 8.0 | 3.0 | 13.0 | 2.6 | 4.0 | 84.4 | 88.6 |
| BR_EV [a] | 1421.0 | 169.0 | 61.6 | 20.5 | 28.8 | 41.0 | 22.5 | 22.5 | 365.8 | 1786.8 |
| BR_DE [a] | 7442.0 | 806.6 | 426.0 | 97.8 | 179.7 | 262.4 | 100.8 | 60.2 | 1934.0 | 9376.0 |
| Shrub | 2295.0 | 183.3 | 132.7 | 89.0 | 70.9 | 89.0 | 42.9 | 30.6 | 638.3 | 2933.3 |
| Grass | 188.1 | 2.2 | 2.2 | 0.3 | 1.8 | 1.0 | 0.7 | 0.2 | 8.4 | 196.5 |
| Crop | 119.4 | 9.2 | 9.9 | 2.0 | 7.4 | 4.6 | 3.2 | 1.4 | 37.8 | 157.2 |

[a]: Refer to Table 1;
[b]: T_MT: Total monoterpenes emission budget of each PFT, T_ALL: Total BVOC emission budget of each PFT, including isoprene and
monoterpenes.



**Table 6 Comparisons of the estimated BVOC budgets (Tg C yr⁻¹) with previous studies**

| Base year | Emission Budget | | | Reference |
|---|---|---|---|---|
| | ISO [a] | MT [a] | Total | |
| 2018 | 11.5 | 3.2 | 14.7 | This study |
| - | 15.0 | 4.3 | 19.3 | Guenther et al. (1995) [a] |
| - | 4.1 | 3.5 | 7.6 | Klinger et al. (2002) [a] |
| 2004 | 6.8 | 2.8 | 9.6 | Tie et al. (2006) [b] |
| 2000 | 10.0 | 2.5 | 12.5 | Guenther et al. (2006) [c] |
| Averaged over 2001-2006 | 9.6 | 2.8 | 12.4 | Fu and Liao (2012) [c] |
| - | 12.7 | | | Fu et al. (2007) [d] |
| 2012 | 5.7 | | | Stavrakou et al. (2014) [d] |

[a]: Based on G95 algorithms (Guenther et al., 1995);
[b]: Based on G20 algorithms (Guenther et al., 2000);
[c]: Based on MEGAN2.0 algorithms (Guenther et al., 2006);
[d]: Top-down annual emission estimates which were inferred by inversion of GOME formaldehyde columns.



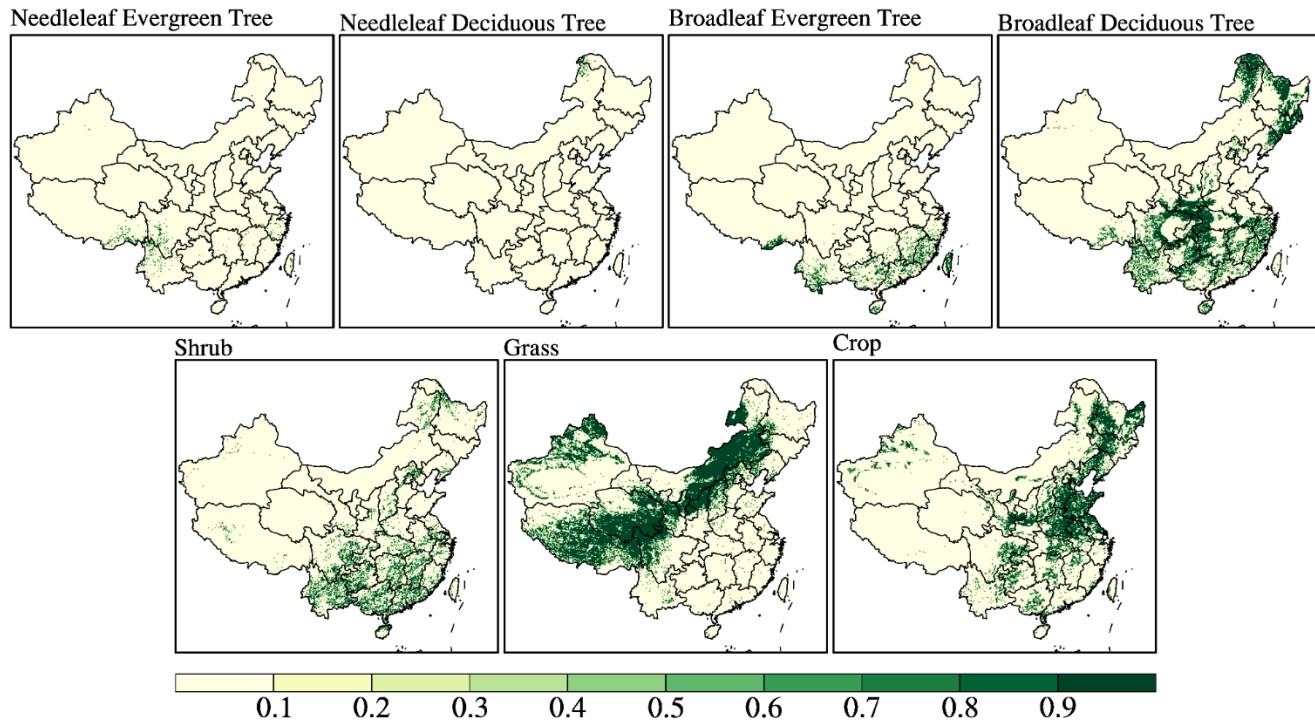

Figure 1 Spatial distribution of dominant plant functional types (PFTs) in China.





**Figure 2 Season-averaged simulated air temperature at 2 m (T2) (left panel), simulated downward solar radiation (SWDOWN) (middle panel) and LAI distributions (right panel) in China.**





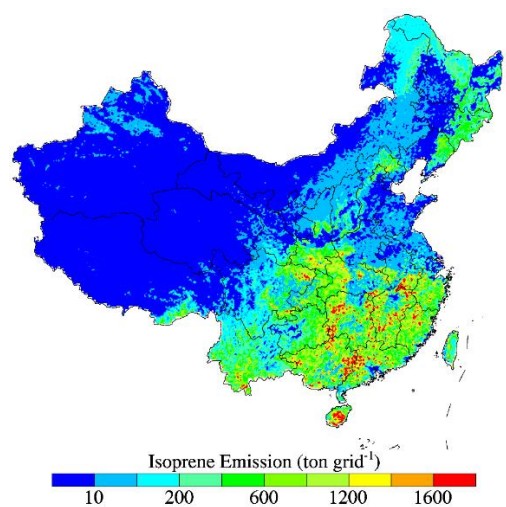

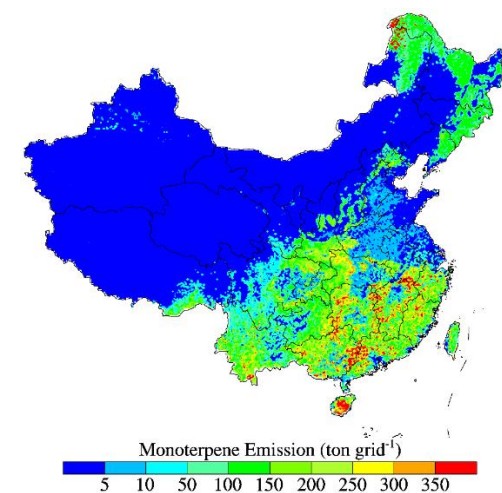

**Figure 3 Spatial distribution of isoprene and monoterpenes emissions in the year 2018.**

(a)  (b)

(c)  (d)

Isoprene Emission Flux (g C km$^{-2}$ h$^{-1}$)

10  200  600  1000  3000

517

**Figure 4 Average seasonal isoprene emission fluxes (g C km$^{-2}$ h$^{-1}$): (a) spring; (b) summer; (c) autumn; (d) winter.**





(a)

(b)

(c)

(d)

Monoterpene Emission Flux (g C km$^{-2}$ h$^{-1}$)

10   50   150   250   350   450   550   650

519

**Figure 5 Average seasonal monoterpene emission fluxes (g C km$^{-2}$ h$^{-1}$): (a) spring; (b) summer; (c) autumn; (d) winter.**





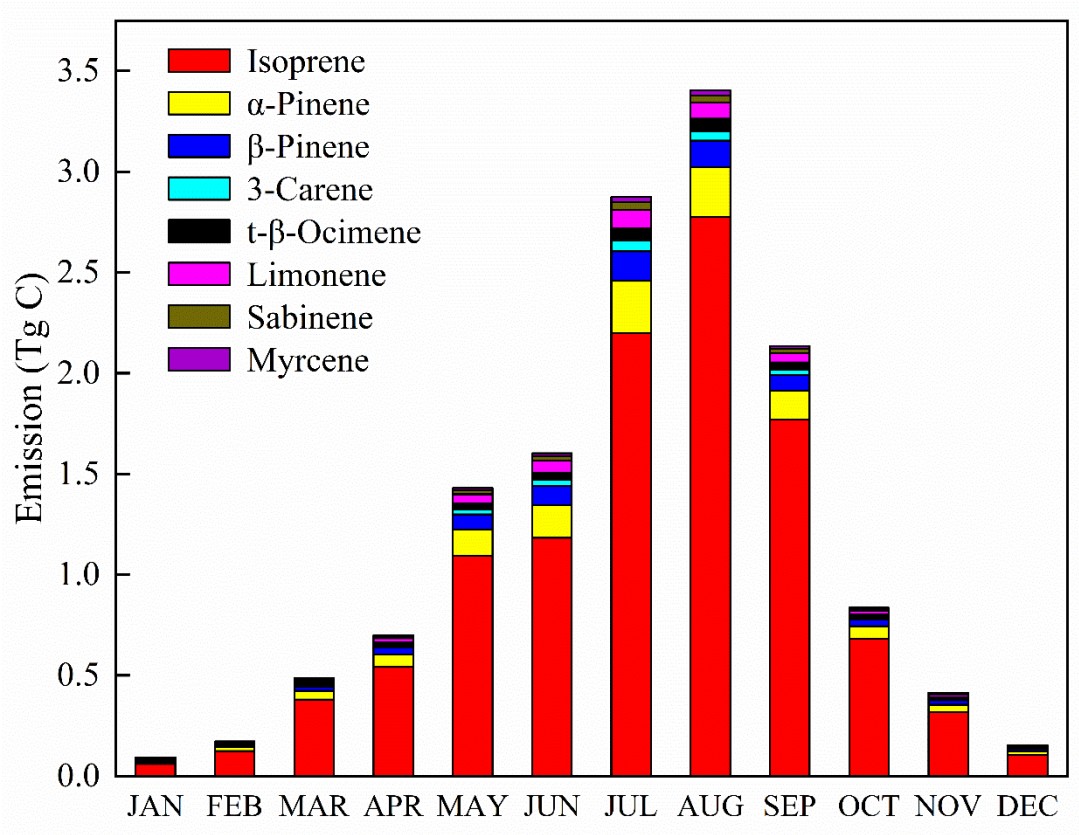

521

Figure 6    Calculated monthly variation of BVOC emission budget (Tg C).

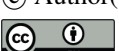
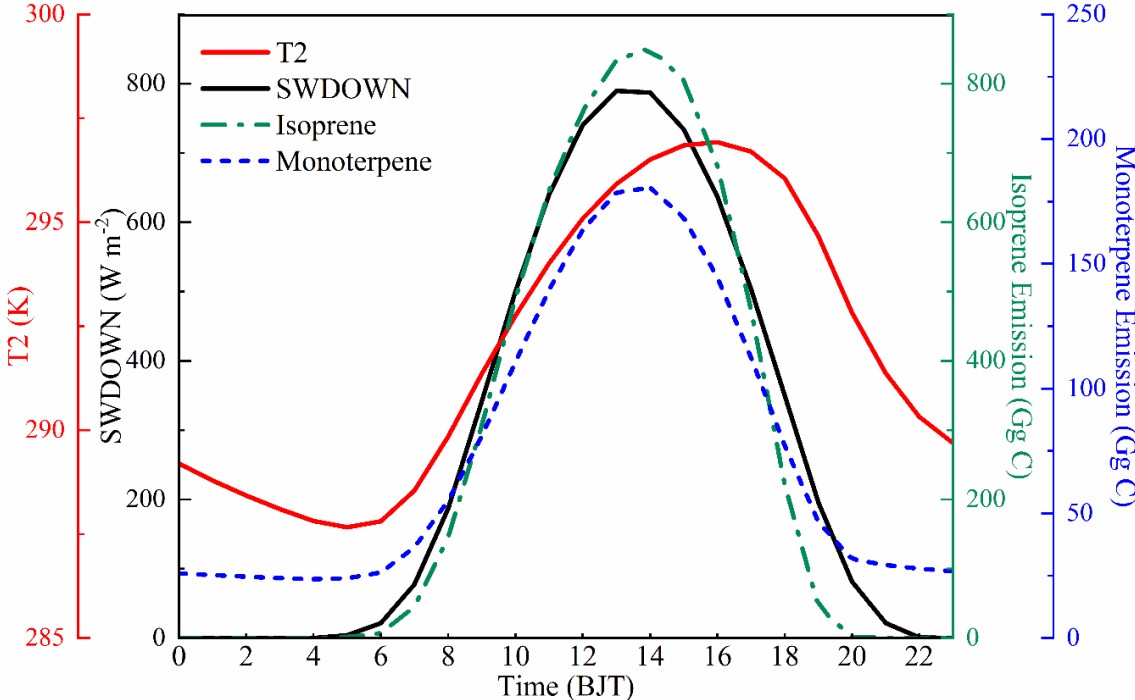

523
**Figure 7 Diurnal variations in summer of temperature at 2 m height (T2), downward solar radiation (SWDOWN) and emissions of**
**isoprene and monoterpenes (Gg C); BJT: Beijing Time.**