# Peer review of "Estimation of biogenic volatile organic compound (BVOC) emissions in China using WRF-CLM-MEGAN coupled model"

_Biogeosciences, 2019_

## Referee Comment (RC1) · Anonymous Referee #1 · 23 Jan 2020

Yin et al present an emissions inventory for some BVOC across China, using the BVOC emissions model Megan and WRF-CLM. A number of previous studies have already presented emission inventories for China, as also mentioned by the authors. While their approach is somewhat different, I cannot get overly excited about the analysis as it is currently presented. It seems methodologically largely sound (but see my questions below); but apart from providing yet another BVOC emissions map there is little novelty in the study. Isoprene and MT emissions are largest from forests, most of the MT come from conifers; spatial patterns thus depend on weather as well as land cover, temporal patterns on weather and LAI — all of this has been found in many other studies before and the results are no surprise given the main BVOC algorithms that basically

vary standard emission factors (which depend on the PFT) with weather.

I would think that at least some simulation experiments that investigate emission changes historically and/or in future, in response to climate change, CO2 and/or land use change would be needed to warrant publication of the work. Given that the model system is in place this should not be too much effort and would make the paper somewhat more interesting.

While I agree that high spatial resolution is an advantage when estimating BVOC emissions I do not follow the need to use a coupled WRF-CLM model version. This would have been needed if the authors wanted to do surface-atmosphere feedback experiments. But at the moment, all they do is to drive an BVOC emissions model with simulated high-resolution weather. This could have been done in offline experiments just as well, especially since the vegetation in this study is prescribed from MODIS.

Lines 68-70: This statement is plain wrong. There are several approaches published in which authors have incorporated BVOC emission algorithms into ecosystem models that calculate also leaf gas exchange, and vegetation dynamics. See e.g. Unger et al., ACP 2013; Pacifico et al., ACP 2011, doi:10.5194/acp-11-4371-2011; Arneth et al., ACP 2007; Schurgers et al., Biogeosc. 2009. And there's presumably others. The authors will need to do their homework more thoroughly. And since the authors didn't even use the dynamic vegetation module of CLM but prescribed land cover and LAI their claim that the study presented here is so much more 'accurate' (see line 79) than previous estimates seems overstated.

MEGAN defines canopy-level emission factors for multiple compounds. What is the observational evidence, if any, for compounds such as myrcene, ocimene, sabinene etc? Where and what types of have canopy-scale measurements been made that would in fact support the value specified? And if such measurements are scarce/nonexistent, what justifies their use in a large-scale inventory?

Guenther et al. in their 2012 paper claim that they describe how leaf age fraction is

estimated in section 2.4 of their paper, but section 2.4 has no mentioning of leaf age calculation at all (neither has any other section in their manuscript). The authors need to describe how leaf age (new, growing, mature and senescing categories) can be differentiated (realistically) from a remote sensing product such as MODIS.

Worden et al (ACP, 2019, https://doi.org/10.5194/acp-19-13569-2019) use MOPITT CO to infer isoprene emissions. While they don't present numbers for China, I wonder if it would not be worthwhile to approach these authors to find out if an emission map for China could be obtained in order to compare with the simulations shown here.

The section 'uncertainties' is somewhat thin. Nothing in it is wrong but it doesn't provide a lot of substance and the section reads a bit as an after-thought at the moment. Needs more concrete examples and/or even example simulations. And the section 'conclusions' is in fact merely a summary section, which could be removed, as the text contains a lot of repetition of what has been said elsewhere in the manuscript.

Scientific papers should be apolitical. Yet I note that Taiwan (Republic of China) is shown on the maps presented. Moreover, while not all of the islands in the South China Sea are shown on these maps (the Paracel islands and Pratas seem to be included) there are several occasions in the paper in which the authors state 'the small islands in the South China Sea are not included', implying that these should be counted. These islands are contested territory and the political status of Taiwan is also a non-consensual one. The authors write about emissions from China, and not emissions from the People's Republic of China, but in the day-to-day use of China the term is synonymous with PRC. I am concerned that misunderstanding might arise from this, and therefore suggest to restrict the reporting of numbers and maps to mainland China only.

A number of more minor issues: Line 36 – sounds as if global BVOC emissions are known to be 1150 Tg C a-1, which is not the case. There are huge uncertainties and no global observations. Revise. Line 50 '. . .that cannot be reproduced well in canopy

environment models'. Unclear what is a canopy environment model. Lines 62, 66 and possibly elsewhere: hydrological cycle (not hydrologic cycle).

---

## Referee Comment (RC2) · Anonymous Referee #2 · 31 Jan 2020

BVOCs emission shows great impact on atmospheric chemistry and global climate due to its high chemical reactivity and high loads. This study uses a WRF-CLM-MEGAN coupled model to simulate BVOCs emission over China by update the input data. The paper is well written and the structure is well organized. But the key issue is, the paper is lack of innovation. Since a series of previous studies have been conducted to estimate BVOCs emission over China or other regions. The author mentioned that the CLM is originally coupled with earth climate models and the spatial resolution is coarse. This study embedded MEGAN model within WRF-CLM with higher spatial resolution, and improved the input data by using satellite data. But higher resolution and high quality of input data is insufficient, since many previous studies had also updated the input

data. The author also mentioned that the processes of land biogeophysics, hydrologic cycle, biogeochemistry etc have great impact of BVOCs estimation, but compare with input data, which one is more important on the estimation of BVOCs? Which one has greater uncertainties. Does the performance of BVOCs simulation improve by considering the land surface processes comparing with the BVOCs simulation without the consideration of land surface processes. Furthermore, the spatial and temporal variations of BVOCs over China are quite clear in previous studies. Therefore, what's the main differences and new findings compare with previous studies?

---

## Referee Comment (RC3) · Anonymous Referee #3 · 7 Feb 2020

General comments: This paper focused on estimating BVOC emissions in China for the year 2018 by using WRF-CLM-MEGAN. My main concern is about the 'new' knowledge brought up by this paper. The paper used the coupled version of model, but the current results are insufficient to present new knowledge from this simulation or the benefits of using WRF + CLM set up for running MEGAN. From the spatial and temporal patterns, we cannot simply see where the improvements are in comparation with previous estimations. Furthermore, the evaluation with the observation data compared the modelled result from 2018 with literature values from any another year, which is not rigid at all. The differences in measured and modelled site conditions, such as environmental conditions and plant composition were not discussed or compared in the evaluation,

instead, the authors concluded that we need more measurement data.

The structure of this paper is not well organized. For example, the data information L151 – 154 were placed in the results section; large amount of model description was placed in the introduction part. The aim or hypothesis of this study is not clear.

English needs to improve in this paper.

Detailed comments:

L35-37, please use more recent global estimations.

L45-46, What do you mean with '...as an insolated step outside the whole terrestrial ecosystem processes'? Do you actually mean that BVOC processes are not linked to photosynthesis? Another issue here is that the references the authors listed here were just based on one model. It is not a good idea to conclude this is a general problem for modelling BVOCs only based on one model.

L113-115, it is not clear for me why the authors decided to use MODIS LAI, instead of the modelled LAI and dynamic vegetation, and also why not consider CO2 impacts on emissions? Please clarify. MODIS LAI is 8 days intervals, how the author deal with the days between 2 LAI images. Another issue is that: did the authors consider quality flag for the LAI product?

L121, "... and each run covered 31 days ..." where these runs refer to? Please clarify.

L148, not clear for me what modification/process have been implemented.

---

## Author Comment (AC1) · 30 Apr 2020

**Response to referee comment #1:**

*Yin et al present an emissions inventory for some BVOC across China, using the BVOC emissions model Megan and WRF-CLM. A number of previous studies have already presented emission inventories for China, as also mentioned by the authors. While their approach is somewhat different, I cannot get overly excited about the analysis as it is currently presented. It seems methodologically largely sound (but see my questions below); but apart from providing yet another BVOC emissions map there is little novelty in the study. Isoprene and MT emissions are largest from forests, most of the MT come from conifers; spatial patterns thus depend on weather as well as land cover, temporal patterns on weather and LAI;all of this has been found in many other studies before and the results are no surprise given the main BVOC algorithms that basically vary standard emission factors (which depend on the PFT) with weather.*

**Response:** The manuscript was revised much according to three referees' comments.

The novelty in this study is that the BVOCs emission is estimated by including some PFT-specific physiological parameters. These parameters are derived from CLM4, but never considered in the previous BVOC estimation algorithms coupled in the weather forecasting models.

We found the improvements are important (more details could be found in the section 3.2). Firstly, the estimations by using leaf temperature in our study were about 20 % higher than those estimated based on air temperature as in the previous methods. Secondly, the separate treatments of sunlit and shaded leaves in this study, which affect within canopy solar radiation, lowered the estimations by a factor of 2 compared with estimates made by methods neglecting shaded canopy. Thirdly, in this study, leaf temperature and solar radiation were averaged over the past running time at each time step to estimate emission response to weather history. However, in the original code, this response was estimated based on fixed parameters. The improved representation in our study resulted in 50 % higher estimations than those based on fixed values.

The results were within a factor of 2 of most canopy-scale flux measurements and top-down isoprene inventories, indicating an overall good performance of the coupled model (section 4).

*I would think that at least some simulation experiments that investigate emission changes historically and/or in future, in response to climate change, CO2 and/or land use change would be needed to warrant publication of the work. Given that the model system is in place this should not be too much effort and would make the paper somewhat more interesting.*

**Response:** The primary purpose of this study is to improve BVOC estimation in weather forecasting models by using physiological parameters provided by CLM4 land surface scheme. New discussions have been added in the section 3.2 (Page 7, Line 201) of the revised manuscript.

*While I agree that high spatial resolution is an advantage when estimating BVOC emissions I do not follow the need to use a coupled WRF-CLM model version. This would have been needed if the authors wanted to do surface-atmosphere feedback experiments. But at the moment, all they do is to drive an BVOC emissions model with simulated high-resolution weather. This could have been done in offline experiments just as well, especially since the vegetation in this study is prescribed from MODIS.*

**Response:** We reworded the Introduction section to illustrate the reason why we use CLM4 land surface scheme. The CLM4 scheme was used to provide real-time vegetation physiological factors through the parameterization of comprehensive ecological and biological processes for MEGAN. Related factors have been described in the Method section. The impacts of reasonable physiological parameter applications on estimates were discussed in section 3.2 of the revised manuscript.

**Revisions:** (Page 2, Line 44) "It has been demonstrated that BVOC emission rates are heavily affected by plant physiological factors and environmental conditions (Peñuelas and Staudt, 2010). Many biogenic emission models have been developed with a strong foundation in the physiological processes of a leaf (Guenther et al., 1991; Niinemets et al., 1999; Martin et al., 2000). The Model of Emissions of Gases and Aerosols from Nature (MEGAN) that estimates BVOC emission fluxes as basal emission rates modulated by emission activity factors has been intensively used for regional and global BVOC emission estimations (Guenther et al., 2006; Guenther et al., 2012). Process-based models linked BVOC production rate explicitly to leaf photosynthetic electron transport rate and election requirement for BVOC synthesis (Niinemets et al., 1999; Niinemets et al., 2002).

The MEGAN algorithms have been incorporated into Community Land Model (CLM), the terrestrial component of the earth climate system model, as one step toward integrating biogeochemical processes in the model. In the coupling of MEGAN and CLM, all the physical and biological variables required by BVOC estimation are determined by comprehensive ecological and physiological processes parameterized in CLM at each time step (Levis et al., 2003; Oleson et al., 2010; Lawrence et al., 2011). Process-based models are typically coupled within dynamic vegetation models that have a mechanistic model for leaf photosynthesis at their core (Arneth et al., 2007; Pacifico et al., 2011; Yue and Unger, 2015). In general, these coupled models are employed to investigate the long-term interactions and feedbacks between terrestrial vegetation and climate change with spin-up and simulation time from tens to thousands of years.

Instead of coupling detailed algorithms within the land surface parameterizations, a simplified version of MEGAN algorithm, the parameterized canopy emission activity (PCEEA) algorithm, has been coupled with weather and climate forecasting models as an independent module to generate online biogenic emission inventory for atmospheric chemistry simulation (Guenther et al., 2006; Sakulyanontvittaya et al., 2008;

Fu and Liao, 2012; Henrot et al., 2017). Instead of using a detailed canopy model to calculate leaf temperature and leaf-level photosynthetic photo flux density (PPFD), the PCEEA algorithm parameterizes the modification of these plant physiological variables on emission rates based on ambient temperature and canopy above solar radiation. Although leaf temperature is strongly related to ambient temperature, it is also affected by other physical and biological factors such as irradiation and evapotranspiration. Subin et al. (2011) indicated that the strong advection and boundary layer mixing during the day decoupled the air temperature from the vegetation temperature to a great extent, making daytime surface energy budget the primary controlling factors of vegetation temperature changes. Furthermore, due to the different morphological and physiological properties, relationships between air temperature and leaf temperature, and between canopy above PPFD and leaf-level PPFD, vary significantly among tree species. Since the PCEEA algorithm was based on standard MEGAN canopy model simulations for warm broadleaf forests, using the same equations for representations of other plant types leads to unpredictable uncertainties. Leaf temperature and PPFD averaged over the past 24 and 240 h are used in MEGAN algorithms to account for effects of medium-term weather history. However, the PCEEA algorithm obtains the past conditions from a prescribed climatological monthly mean dataset, which could be much different from the real meteorology (Zhao et al., 2016). Therefore, reasonable plant-specific physiological variables are needed to improve the BVOC estimation in weather models.

CLM version 4 (CLM4) was coupled and released with the Weather Research and Forecasting model (WRF), a mesoscale numerical model designed to simulate regional weather and climate, since version 3.5 as one of the land surface scheme options to better characterize land surface processes (Jin and Wen, 2012; Jin et al., 2010; Subin et al., 2011). Because MEGAN has been embedded within CLM as mentioned above, the coupling of WRF-CLM4-MEGAN allowed regional weather forecasting models to estimate BVOC emissions within a comprehensive ecological climatology framework. Besides improvements result from real-time plant physiological variables derived from land surface parameterizations, sub-grid vegetation compositions represented in CLM4 are also expected to provide a more reasonable estimation in view of the significant variability in basal emission ability among tree species. However, few studies employed the coupled mode to estimate regional BVOC emissions (Zhao et al., 2016)."

*Lines 68-70: This statement is plain wrong. There are several approaches published in which authors have incorporated BVOC emission algorithms into ecosystem models that calculate also leaf gas exchange, and vegetation dynamics. See e.g. Unger et al., ACP 2013; Pacifico et al., ACP 2011, doi:10.5194/acp-11-4371-2011; Arneth et al., ACP 2007; Schurgers et al., Biogeosc. 2009. And there's presumably others. The*

*authors will need to do their homework more thoroughly. And since the authors didn't even use the dynamic vegetation module of CLM but prescribed land cover and LAI their claim that the study presented here is so much more 'accurate' (see line 79) than previous estimates seems overstated.*

**Response:** We reworded the Introduction section. Most studies have integrated BVOC emission algorithms into land surface and dynamic vegetation models to investigate the response of vegetation distributions and emissions to long-term climate change. The purpose of our study is to use physiological factors provided by CLM4 to improve estimates of regional BVOC emissions in weather and climate forecasting models. The significant effects of reasonable physiological parameter applications on estimates were investigated.

**Revisions:** (Page 2, Line 51) "The MEGAN algorithms have been incorporated into Community Land Model (CLM), the terrestrial component of the earth climate system model, as one step toward integrating biogeochemical processes in the model. In the coupling of MEGAN and CLM, all the physical and biological variables required by BVOC estimation are determined by comprehensive ecological and physiological processes parameterized in CLM at each time step (Levis et al., 2003; Oleson et al., 2010; Lawrence et al., 2011). Process-based models are typically coupled within dynamic vegetation models that have a mechanistic model for leaf photosynthesis at their core (Arneth et al., 2007; Pacifico et al., 2011; Yue and Unger, 2015). In general, these coupled models are employed to investigate the long-term interactions and feedbacks between terrestrial vegetation and climate change with spin-up and simulation time from tens to thousands of years."

(Page 13, Line 387) "This framework improved biogenic emission estimations by using reasonable PFT-specific physiological parameters derived from CLM4 land surface scheme to drive standard MEGAN algorithms. The simulated leaf temperature was typically higher than air temperature by 1~6 K in daytime and lower than ambient value by approximately 2 K during night. Using air temperature to parameterize BVOC emissions underestimated isoprene and monoterpene emissions in July by 23.9 % and 21.9 %, respectively. Because the sunlit fraction of broadleaf trees was typically lower than 0.3 in July, ignoring the difference in absorbed radiation of sunlit and shaded leaves overestimated emissions from broadleaf trees by a factor of 2.7. Assigning fixed values to variables that related to previous conditions made a similar estimation in January with using dynamic variables, while underestimated emissions in July by approximately 50 %. Due to the significant discrepancy caused by differences in driving variables, more reasonable parameter applications are important for accurately estimating biogenic emissions."

*MEGAN defines canopy-level emission factors for multiple compounds. What is the*

*observational evidence, if any, for compounds such as myrcene, ocimene, sabinene etc?*
*Where and what types of have canopy-scale measurements been made that would in*
*fact support the value specified? And if such measurements are scarce/non-existent,*
*what justifies their use in a large-scale inventory?*

**Response:** MEGAN defines emission factors based on about 300 studies. Although
most studies measured emission rates of the sum of monoterpenes or of
several abundant compounds such as α-pinene and β-pinene, a number of
studies reported emission rates of myrcene, ocimene, sabinene based on
enclosure or canopy-scale measurements, such as Geron et al. (2000);
Harrison et al. (2001); Kim (2001); Dindorf et al. (2006); Holzke et al.
(2006); Bai et al. (2012) etc.. Compared with studies based on enclosure
method, much less studies offering emission rates of the above monoterpene
species were based on canopy-scale methods. More research on these
compounds are needed.

In our study, the observed isoprene emission factors in China, as well as
those used in previous studies were used to determine isoprene emission
factors. References were listed in the Supplement Table 1. Due to the scarce
reports of local monoterpene emissions factor, emission factors of
monoterpene species were roughly estimated based on global average
emissions factors suggested in Guenther et al. (2012) and modified by the
ratio of local to global isoprene emission factors. Emission factors is a
significant source of uncertainty due to the difficulties in conducting large-
scale and long-term measurements, which has been recognized in the
Uncertainty section.

**Revisions:** (Page 4, Line 107) "PFT-specific EFs of isoprene were determined based
on observations conducted in China and EF used in previous studies (as
shown in Table S1). Due to the lack of detailed monoterpene EFs reports,
the EFs of main monoterpene species were determined by scaling default
MEGAN EFs with the ratio of local isoprene EF to default value presented
in Guenther et al. (2012). Table 1 summarizes the EFs for each vegetation
type used in this study."

*Guenther et al. in their 2012 paper claim that they describe how leaf age fraction is*
*estimated in section 2.4 of their paper, but section 2.4 has no mentioning of leaf age*
*calculation at all (neither has any other section in their manuscript). The authors need*
*to describe how leaf age (new, growing, mature and senescing categories) can be*
*differentiated (realistically) from a remote sensing product such as MODIS.*

**Response:** Accepted. Details of emission activity algorithms can be found in Guenther
et al. (2006). The determination of leaf age fraction was described in the
section 3.2.2 of their paper. The division of leaf age fraction is based on the
change in LAI between the present time step and the previous time step. The
MODIS 8-day LAI data was used in our study. Every 8 days used the same
MODIS LAI database and LAI of the previous time step was set as the
previous 8-day MODIS LAI data. We added the reference and brief

descriptions in our Methods and Data section.

**Revisions:** (Page 4, Line 112) "The emission activity factor for each compound ($\gamma_i$) accounts for emission responses to solar radiation, leaf temperature, LAI, leaf age and soil moisture. The effects of variations in $CO_2$ concentration was neglected in this study because the simulation was performed for only one year and Heald et al. (2009) found that accounting for $CO_2$ inhibition has litter impact on predictions of present-day isoprene emission. Details of the algorithms could be found in Guenther et al. (2006) and Guenther et al. (2012)."

(Page 6, Line 167) "The same LAI data was used as current LAI ($LAI_c$) for 8 days and the past 8-day image was considered as LAI of the previous time step ($LAI_p$). The changes between $LAI_c$ and $LAI_p$ was used to determine leaf age (Guenther et al., 2006)."

*Worden et al (ACP, 2019, https://doi.org/10.5194/acp-19-13569-2019) use MOPITT CO to infer isoprene emissions. While they don't present numbers for China, I wonder if it would not be worthwhile to approach these authors to find out if an emission map for China could be obtained in order to compare with the simulations shown here.*

**Response:** Accepted. Worden et al. (2019) estimated biogenic CO flux based on MOPITT CO observations. They evaluated the seasonal and spatial patterns of the posterior CO against top-down estimates of isoprene fluxes based on OMI formaldehyde observations (Stavrakou et al., 2015; Bauwens et al., 2016). We evaluated our results against the top-down inventory based on OMI observations in the revised manuscript.

**Revisions:** (Page 12, Line 355) "Formaldehyde (HCHO), as a major intermediate product in the degradation of isoprene in the atmosphere, has been widely used as a proxy for estimates isoprene emissions. Fu et al. (2007) used a continuous 6-year record (1996–2001) of Global Ozone Monitoring Experiment (GOME) HCHO columns and made a top-down estimate for isoprene as 12.7 Tg yr$^{-1}$ in China, which is comparable to our model outputs. However, Stavrakou et al. (2014) inferred isoprene emissions by inversion of GOME-2 HCHO columns and the satellite-derived emissions were found to decrease from 8.6 Tg in 2007 to 6.5 Tg in 2012, lower than emissions in this study by 33.7 %–76.9 %. Stavrakou et al. (2015) assessed the consistency of the top-down isoprene fluxes based on GOME-2 and OMI (Ozone Monitoring Instrument) HCHO observations. The isoprene emission from China in 2010 was estimated to be 5.9 Tg with GOM2-2 and 6.5 Tg with OMI data, with large discrepancies in southern China. Aside from the influence of difference meteorological conditions and land cover changes during the past years, the reliability of satellite-based constraints also needs to be improved because that the HCHO is affected by non-isoprene sources plus uncertainty and spatial smearing in isoprene-formaldehyde relationship (Fu et al., 2019)."

*The section 'uncertainties' is somewhat thin. Nothing in it is wrong but it doesn't provide a lot of substance and the section reads a bit as an after-thought at the moment. Needs more concrete examples and/or even example simulations. And the section 'conclusions' is in fact merely a summary section, which could be removed, as the text contains a lot of repetition of what has been said elsewhere in the manuscript.*

**Response:** We revised the Uncertainty and Conclusion section.

**Revisions:** (Page 12, Line 367) "The major areas of uncertainty include basal emission factors, land cover data (vegetation distribution and LAI) and physiological and environmental parameters.

The basal emission factors have been identified as the most important uncertainty source (Zheng et al., 2010; Situ et al., 2014; Wang et al., 2016). Local emission factors for isoprene reported by previous observations conducted in China were used in this study. Since measurements of the monoterpene emission factors are scarce, we calculated local emission factors based on the ratio of local isoprene emission factor to default emission factor in MEGAN literature. There are large uncertainties associated with the conversion approach. More in-situ observations on emission rates of different PFTs in China are required.

The conversion from IGBP land category to PFTs used in CLM4 resulted in uncertainty in vegetation map. For example, broadleaf deciduous trees in CLM4 are group into tropical, temperate and boreal tree categories, with specified canopy parameterizations. The climate zone separations were not considered in our study and all broadleaf deciduous trees in IGBP were grouped into broadleaf deciduous temperate tree category in CLM4. The conversion method may lead to uncertainties when parameterizing trees or shrubs in boreal and tropical area. Wang et al. (2018) adopted three LAI datasets to study the effect of the LAI input on BVOC emissions and indicated that the discrepancies between different LAI inputs do not obviously affect the estimates.

CLM4 parameterizes one layer of canopy, however, solar radiation is attenuated by foliage and leaf temperature varies among layers. A relatively simple representation of canopy is also a source of uncertainty. Guenther et al. (1995) found a less than 5 % difference in global annual isoprene emission estimated with one or five lays and no change in the estimations of other BVOC emissions, suggesting that BVOC emissions are relatively insensitive to the number layers. However, many studies indicated that the treatment of microclimatic factors such as light and leaf temperature within the canopy resulted in substantial difference in estimated emissions (Keenan et al., 2011)."

(Page 13, Line 387) "This study estimated the emission budgets and spatial-temporal patterns of BVOC in China in the year 2018 using the WRF-CLM4-MEGAN modeling system. This framework improved biogenic emission estimations by using reasonable PFT-specific physiological parameters derived from CLM4 land surface scheme to drive standard

MEGAN algorithms. The simulated leaf temperature was typically higher than air temperature by 1~6 K in daytime and lower than ambient value by approximately 2 K during night. Using air temperature to parameterize BVOC emissions underestimated isoprene and monoterpene emissions in July by 23.9 % and 21.9 %, respectively. Because the sunlit fraction of broadleaf trees was typically lower than 0.3 in July, ignoring the difference in absorbed radiation of sunlit and shaded leaves overestimated emissions from broadleaf trees by a factor of 2.7. Assigning fixed values to variables that related to previous conditions made a similar estimation in January with using dynamic variables, while underestimated emissions in July by approximately 50 %. Due to the significant discrepancy caused by differences in driving variables, more reasonable parameter applications are important for accurately estimating biogenic emissions. Using the CLM4-MEGAN framework, the annual emissions of BVOC in China was estimated to be 14.7 Tg C, with isoprene and monoterpenes accounting for 78.3 % and 21.7 % of the totals, respectively. The coupled model successfully reproduced the spatial and temporal patterns of BVOC emissions. Although past studies reported that emissions peaked in July, the highest emission was found in August in our research. This result was consistent with observations, indicating that the past-time leaf temperature and solar radiation must be considered in emission estimation. The predicted values in forest areas during wet seasons were within a factor of 2 of observed values, however, significant discrepancy was found in estimations under drought conditions. The predicted annual emission was within a factor of 2 of top-down estimates and at the upper end of values reported in previous modeling estimates. Comparisons indicated an overall good performance of the model during dominant BVOC emission seasons, but further efforts focusing on improving estimation of emissions in dry areas are still needed."

*Scientific papers should be apolitical. Yet I note that Taiwan (Republic of China) is shown on the maps presented. Moreover, while not all of the islands in the South China Sea are shown on these maps (the Paracel islands and Pratas seem to be in-cluded) there are several occasions in the paper in which the authors state 'the small is-lands in the South China Sea are not included', implying that these should be counted. These islands are contested territory and the political status of Taiwan is also a non-consensual one. The authors write about emissions from China, and not emissions from the People's Republic of China, but in the day-to-day use of China the term is synonymous with PRC. I am concerned that misunderstanding might arise from this, and therefore suggest to restrict the reporting of numbers and maps to mainland China only.*

**Response:** All the coauthors of this paper are Chinese researchers. From our perspective, Taiwan and the South China Sea are shown on the map to ensure that the paper is apolitical. Thank you very much for your

understanding.

*Line 36 – sounds as if global BVOC emissions are known to be 1150 Tg C a-1, which is not the case. There are huge uncertainties and no global observations. Revise.*
**Response:** Accepted.
**Revisions:** (Page 2, Line 37) "Globally speaking, biogenic volatile organic compounds (BVOCs) emitted by terrestrial vegetation are estimated to be 500 ~ 1100 Tg C yr$^{-1}$, corresponding to about 90 % of the emission total (Guenther et al., 1995; Henrot et al., 2017)."

*Line 50 '. . .that cannot be reproduced well in canopy environment models'. Unclear what is a canopy environment model.*
**Response:** We reworded the Introduction section. Canopy environment models typically consist of radiative transfer and energy balance simulations to calculate solar radiation reflected and absorbed by the canopy, its transfer within the canopy, as well as leaf temperature of different layers of the canopy.

*Lines 62, 66 and possibly elsewhere: hydrological cycle (not hydrologic cycle).*
**Response:** Accepted.
**Revisions:** (Page 4, Line 99) "The CLM4 was coupled and released with WRF since version 3.5 as one of the land surface scheme options. CLM4 consists of components related to land biogeophysics, hydrological cycle, biogeochemistry, human dimensions and ecosystem dynamics."

**References:**

Bai, J., Lin, F., Wan, X., Guenther, A., Turnipseed, A., and Duhl, T.: Volatile organic compound emission fluxes from a temperate forest in Changbai Mountain, Acta Scientiae Circumstantiae, 32, 545-554, 2012.

Bauwens, M., Stavrakou, T., Müller, J. F., De Smedt, I., Van Roozendael, M., van der Werf, G. R., Wiedinmyer, C., Kaiser, J. W., Sindelarova, K., and Guenther, A.: Nine years of global hydrocarbon emissions based on source inversion of OMI formaldehyde observations, Atmos. Chem. Phys., 16, 10133-10158, 10.5194/acp-16-10133-2016, 2016.

Dindorf, T., Kuhn, U., Ganzeveld, L., Schebeske, G., Ciccioli, P., Holzke, C., Köble, R., Seufert, G., and Kesselmeier, J.: Significant light and temperature dependent monoterpene emissions from European beech (Fagus sylvatica L.) and their potential impact on the European volatile organic compound budget, Journal of Geophysical Research: Atmospheres, 111, 10.1029/2005jd006751, 2006.

Geron, C., Rasmussen, R., R. Arnts, R., and Guenther, A.: A review and synthesis of monoterpene speciation from forests in the United States, Atmospheric Environment, 34, 1761-1781, https://doi.org/10.1016/S1352-2310(99)00364-7, 2000.

Guenther, A., Karl, T., Harley, P., Wiedinmyer, C., Palmer, P. I., and Geron, C.: Estimates of global terrestrial isoprene emissions using MEGAN (Model of Emissions of Gases and Aerosols from Nature), Atmos. Chem. Phys., 6, 3181-3210, 10.5194/acp-6-3181-2006, 2006.

Harrison, D., Hunter, M. C., Lewis, A. C., Seakins, P. W., Nunes, T. V., and Pio, C. A.: Isoprene and monoterpene emission from the coniferous species Abies Borisii-regis—implications for regional air chemistry in Greece, Atmospheric Environment, 35, 4687-4698, https://doi.org/10.1016/S1352-2310(01)00092-9, 2001.

Holzke, C., Dindorf, T., Kesselmeier, J., Kuhn, U., and Koppmann, R.: Terpene emissions from European beech (shape Fagus sylvatica~L.): Pattern and Emission Behaviour Over two Vegetation Periods, Journal of Atmospheric Chemistry, 55, 81-102, 10.1007/s10874-006-9027-9, 2006.

Kim, J.-C.: Factors controlling natural VOC emissions in a southeastern US pine forest, Atmospheric Environment, 35, 3279-3292, https://doi.org/10.1016/S1352-2310(00)00522-7, 2001.

Stavrakou, T., Müller, J. F., Bauwens, M., De Smedt, I., Van Roozendael, M., De Mazière, M., Vigouroux, C., Hendrick, F., George, M., Clerbaux, C., Coheur, P. F., and Guenther, A.: How consistent are top-down hydrocarbon emissions based on formaldehyde observations from GOME-2 and OMI?, Atmos. Chem. Phys., 15, 11861-11884, 10.5194/acp-15-11861-2015, 2015.

---

## Author Comment (AC2) · 30 Apr 2020

**Response to referee comment #2:**

*BVOCs emission shows great impact on atmospheric chemistry and global climate due to its high chemical reactivity and high loads. This study uses a WRF-CLM-MEGAN coupled model to simulate BVOCs emission over China by update the input data. The paper is well written and the structure is well organized.*

*But the key issue is, the paper is lack of innovation. Since a series of previous studies have been conducted to estimate BVOCs emission over China or other regions. The author mentioned that the CLM is originally coupled with earth climate models and the spatial resolution is coarse. This study embedded MEGAN model within WRF-CLM with higher spatial resolution, and improved the input data by using satellite data. But higher resolution and high quality of input data is insufficient, since many previous studies had also updated the input data.*

**Response:** The manuscript was revised much according to three referees' comments.

The novelty in this study is that the BVOCs emission is estimated by including some PFT-specific physiological parameters. These parameters are derived from CLM4, but never considered in the previous BVOC estimation algorithms coupled in the weather forecasting models.

We found the improvements are important (more details could be found in the section 3.2). Firstly, the estimations by using leaf temperature in our study were about 20 % higher than those estimated based on air temperature as in the previous methods. Secondly, the separate treatments of sunlit and shaded leaves in this study, which affect within canopy solar radiation, lowered the estimations by a factor of 2 compared with estimates made by methods neglecting shaded canopy. Thirdly, in this study, leaf temperature and solar radiation were averaged over the past running time at each time step to estimate emission response to weather history. However, in the original code, this response was estimated based on fixed parameters. The improved representation in our study resulted in 50 % higher estimations than those based on fixed values.

The results were within a factor of 2 of most canopy-scale flux measurements and top-down isoprene inventories, indicating an overall good performance of the coupled model (section 4).

*The author also mentioned that the processes of land biogeophysics, hydrologic cycle, biogeochemistry etc have great impact of BVOCs estimation, but compare with input data, which one is more important on the estimation of BVOCs? Which one has greater uncertainties. Does the performance of BVOCs simulation improve by considering the land surface processes comparing with the BVOCs simulation without the consideration of land surface processes.*

**Response:** The land surface processes in CLM4 were used to provide real-time plant physiological parameters for MEGAN algorithms. Additional experiments were performed to investigate the influence of considering land surface processes on estimations. Details of results and discussions were presented

in the section 3.2 (Page 7, Line 201) of the revised manuscript.

*Furthermore, the spatial and temporal variations of BVOCs over China are quite clear in previous studies. Therefore, what's the main differences and new findings compare with previous studies?*

**Response:** We reworded the Introduction section to clarify the novelty of our study. The CLM4 scheme was used to provide real-time vegetation physiological factors through the parameterization of comprehensive ecological and biological processes for MEGAN, while most estimates made by weather forecasting models were based on ambient environmental factors. The impacts of physiological parameter applications on estimates were investigated in additional modeling experiments and discussed in section 3.2 of the revised manuscript.

**Revisions:** (Page 2, Line 51) "The MEGAN algorithms have been incorporated into Community Land Model (CLM), the terrestrial component of the earth climate system model, as one step toward integrating biogeochemical processes in the model. In the coupling of MEGAN and CLM, all the physical and biological variables required by BVOC estimation are determined by comprehensive ecological and physiological processes parameterized in CLM at each time step (Levis et al., 2003; Oleson et al., 2010; Lawrence et al., 2011). Process-based models are typically coupled within dynamic vegetation models that have a mechanistic model for leaf photosynthesis at their core (Arneth et al., 2007; Pacifico et al., 2011; Yue and Unger, 2015). In general, these coupled models are employed to investigate the long-term interactions and feedbacks between terrestrial vegetation and climate change with spin-up and simulation time from tens to thousands of years.

Instead of coupling detailed algorithms within the land surface parameterizations, a simplified version of MEGAN algorithm, the parameterized canopy emission activity (PCEEA) algorithm, has been coupled with weather and climate forecasting models as an independent module to generate online biogenic emission inventory for atmospheric chemistry simulation (Guenther et al., 2006; Sakulyanontvittaya et al., 2008; Fu and Liao, 2012; Henrot et al., 2017). Instead of using a detailed canopy model to calculate leaf temperature and leaf-level photosynthetic photo flux density (PPFD), the PCEEA algorithm parameterizes the modification of these plant physiological variables on emission rates based on ambient temperature and canopy above solar radiation. Although leaf temperature is strongly related to ambient temperature, it is also affected by other physical and biological factors such as irradiation and evapotranspiration. Subin et al. (2011) indicated that the strong advection and boundary layer mixing during the day decoupled the air temperature from the vegetation temperature to a great extent, making daytime surface energy budget the primary controlling factors of vegetation temperature changes. Furthermore,

due to the different morphological and physiological properties, relationships between air temperature and leaf temperature, and between canopy above PPFD and leaf-level PPFD, vary significantly among tree species. Since the PCEEA algorithm was based on standard MEGAN canopy model simulations for warm broadleaf forests, using the same equations for representations of other plant types leads to unpredictable uncertainties. Leaf temperature and PPFD averaged over the past 24 and 240 h are used in MEGAN algorithms to account for effects of medium-term weather history. However, the PCEEA algorithm obtains the past conditions from a prescribed climatological monthly mean dataset, which could be much different from the real meteorology (Zhao et al., 2016). Therefore, reasonable plant-specific physiological variables are needed to improve the BVOC estimation in weather models.

CLM version 4 (CLM4) was coupled and released with the Weather Research and Forecasting model (WRF), a mesoscale numerical model designed to simulate regional weather and climate, since version 3.5 as one of the land surface scheme options to better characterize land surface processes (Jin and Wen, 2012; Jin et al., 2010; Subin et al., 2011). Because MEGAN has been embedded within CLM as mentioned above, the coupling of WRF-CLM4-MEGAN allowed regional weather forecasting models to estimate BVOC emissions within a comprehensive ecological climatology framework. Besides improvements result from real-time plant physiological variables derived from land surface parameterizations, sub-grid vegetation compositions represented in CLM4 are also expected to provide a more reasonable estimation in view of the significant variability in basal emission ability among tree species. However, few studies employed the coupled mode to estimate regional BVOC emissions (Zhao et al., 2016)."

---

## Author Comment (AC3) · 30 Apr 2020

**Response to referee comment #3:**

*This paper focused on estimating BVOC emissions in China for the year 2018 by using WRF-CLM-MEGAN. My main concern is about the 'new' knowledge brought up by this paper. The paper used the coupled version of model, but the current results are insufficient to present new knowledge from this simulation or the benefits of using WRF + CLM set up for running MEGAN. From the spatial and temporal pat-terns, we cannot simply see where the improvements are in comparation with previous estimations.*

**Response:** The manuscript was revised much according to three referees' comments.

The novelty in this study is that the BVOCs emission is estimated by including some PFT-specific physiological parameters. These parameters are derived from CLM4, but never considered in the previous BVOC estimation algorithms coupled in the weather forecasting models.

We found the improvements are important (more details could be found in the section 3.2). Firstly, the estimations by using leaf temperature in our study were about 20 % higher than those estimated based on air temperature as in the previous methods. Secondly, the separate treatments of sunlit and shaded leaves in this study, which affect within canopy solar radiation, lowered the estimations by a factor of 2 compared with estimates made by methods neglecting shaded canopy. Thirdly, in this study, leaf temperature and solar radiation were averaged over the past running time at each time step to estimate emission response to weather history. However, in the original code, this response was estimated based on fixed parameters. The improved representation in our study resulted in 50 % higher estimations than those based on fixed values.

The results were within a factor of 2 of most canopy-scale flux measurements and top-down isoprene inventories, indicating an overall good performance of the coupled model (section 4).

*Furthermore, the evaluation with the observation data compared the modelled result from 2018 with literature values from any another year, which is not rigid at all. The differences in measured and modelled site conditions, such as environmental conditions and plant composition were not discussed or compared in the evaluation, instead, the authors concluded that we need more measurement data.*

**Response:** Accepted. The monthly average or total emissions were used for comparison to minimize the influence of short-term differences in meteorological conditions. The differences in measured and modelled site conditions were also included in the revised manuscript.

**Revisions:** (Page 11, Line 329) "We evaluated the inventory against canopy-scale measurements conducted at different sites in China. Given that years of interest in the present study and observations are different, the average data or monthly total emissions were used for comparison.

Using eddy covariance technique, Baker et al. (2005) measured isoprene fluxes in Xishuangbanna, Yunnan Province (21.92°N, 101.27°E). Daytime

isoprene fluxes during the wet season (July 2002) was approximately 1 mg C m$^{-2}$ h$^{-1}$. Dry season (February and March) daytime fluxes averaged about 0.15 mg C m$^{-2}$ h$^{-1}$ and maximum fluxes were over 0.6 mg C m$^{-2}$ h$^{-1}$. Our model predicted a similar daytime average isoprene flux in wet season as 1.5 mg C m$^{-2}$ h$^{-1}$ at this site. The modeled flux for dry season was 1.19 mg m$^{-2}$ h$^{-1}$, higher than observed maximum value by a factor of 2.

Based on Relaxed Eddy Accumulation (REA) technique, emissions of isoprene and monoterpenes of a temperate forest in Changbai Mountain (42.4°N, 128.1°E) were measured during the summer seasons (June, July, August, September) in 2010 and 2011 (Bai et al., 2015). The mean isoprene emission flux was 0.889 mg m$^{-2}$ h$^{-1}$ and the mean total monoterpene emission flux was 0.143 mg m$^{-2}$ h$^{-1}$, and our results estimated emission fluxes of 1.97 for isoprene and 0.37 mg C m$^{-2}$ h$^{-1}$ for monoterpene. The average PAR and temperature during experimental periods were 837.5 μmol m$^{-2}$ s$^{-1}$ and 22.6 °C, respectively. The simulation resulted in an average PAR of 1160.1 μmol m$^{-2}$ s$^{-1}$ and temperature of 22.23 °C. The average leaf temperature was 23.37 °C. The slight overestimation could be attributed to higher PAR simulated in the model.

Using REA method, Bai et al. (2016) measured emissions from a bamboo (*Phyllostachys violascenes*) plantation in Zhejiang Province (30.3°N, 119.57°E) and the average isoprene emission fluxes were 2.8, 1.1, 0.2, 0.1 mg m$^{-2}$ h$^{-1}$ for the experimental periods in July, August, September and October. The predicted monthly average fluxes of these four months were 2.4, 2.3, 3.2, 0.46 mg m$^{-2}$ h$^{-1}$, respectively. Estimations for July and August were within a factor of 2 of observed values. Large discrepancies were found in September and November. The comparisons indicated an overall good performance of the WRF-CLM4-MEGAN in forest areas during wet seasons. However, the large difference associated with estimations for dry areas and seasons clearly suggested that additional investigation and improvements are needed."

*The structure of this paper is not well organized. For example, the data information L151 – 154 were placed in the results section; large amount of model description was placed in the introduction part. The aim or hypothesis of this study is not clear.*
**Response:** Accepted. We rewrote the Introduction and Method and Data sections.
**Revisions:** (Page 2, Line 51) "The MEGAN algorithms have been incorporated into Community Land Model (CLM), the terrestrial component of the earth climate system model, as one step toward integrating biogeochemical processes in the model. In the coupling of MEGAN and CLM, all the physical and biological variables required by BVOC estimation are determined by comprehensive ecological and physiological processes parameterized in CLM at each time step (Levis et al., 2003; Oleson et al., 2010; Lawrence et al., 2011). Process-based models are typically coupled within dynamic vegetation models that have a mechanistic model for leaf

photosynthesis at their core (Arneth et al., 2007; Pacifico et al., 2011; Yue and Unger, 2015). In general, these coupled models are employed to investigate the long-term interactions and feedbacks between terrestrial vegetation and climate change with spin-up and simulation time from tens to thousands of years.

Instead of coupling detailed algorithms within the land surface parameterizations, a simplified version of MEGAN algorithm, the parameterized canopy emission activity (PCEEA) algorithm, has been coupled with weather and climate forecasting models as an independent module to generate online biogenic emission inventory for atmospheric chemistry simulation (Guenther et al., 2006; Sakulyanontvittaya et al., 2008; Fu and Liao, 2012; Henrot et al., 2017). Instead of using a detailed canopy model to calculate leaf temperature and leaf-level photosynthetic photo flux density (PPFD), the PCEEA algorithm parameterizes the modification of these plant physiological variables on emission rates based on ambient temperature and canopy above solar radiation. Although leaf temperature is strongly related to ambient temperature, it is also affected by other ecological conditions such as irradiation and evapotranspiration. Subin et al. (2011) indicated that the strong advection and boundary layer mixing during the day decoupled the air temperature from the vegetation temperature to a great extent, making daytime surface energy budget the primary controlling factors of vegetation temperature changes. Furthermore, due to the different morphological and physiological properties, relationships between air temperature and leaf temperature, and between canopy above PPFD and leaf-level PPFD, vary significantly among tree species. Since the PCEEA algorithm was based on standard MEGAN canopy model simulations for warm broadleaf forests, using the same equations for representations of other plant types leads to unpredictable uncertainties. Leaf temperature and PPFD averaged over the past 24 and 240 h are used in MEGAN algorithm to account for effects of medium-term weather history. However, the PCEEA algorithm obtains the past conditions from a prescribed climatological monthly mean dataset, which could be much different from the real meteorology (Zhao et al., 2016). Therefore, reasonable plant-specific physiological variables are needed to improve the BVOC estimation in weather models.

CLM version 4 (CLM4) was coupled and released with the Weather Research and Forecasting model (WRF), a mesoscale numerical model designed to simulate regional weather and climate, since version 3.5 as one of the land surface scheme options to better characterize land surface processes (Jin and Wen, 2012; Jin et al., 2010; Subin et al., 2011). Because MEGAN has been embedded within CLM as mentioned above, the coupling of WRF-CLM4-MEGAN allowed regional weather forecasting models to estimate BVOC emissions within a comprehensive ecological climatology framework. Besides improvements result from real-time plant physiological

variables derived from land surface parameterizations, sub-grid vegetation compositions represented in CLM4 are also expected to provide a more reasonable estimation in view of the significant variability in basal emission ability among tree species. However, few studies employed the coupled mode to estimate regional BVOC emissions (Zhao et al., 2016)."

*English needs to improve in this paper.*
**Response:** Accepted.

*L35-37, please use more recent global estimations.*
**Response:** Accepted.
**Revisions:** (Page 2, Line 37) "Globally speaking, biogenic volatile organic compounds (BVOCs) emitted by terrestrial vegetation are estimated to be 500 ~ 1100 Tg C yr$^{-1}$, corresponding to about 90 % of the emission total (Guenther et al., 1995; Arneth et al., 2011; Henrot et al., 2017)."

*L45-46, What do you mean with '. . .as an insolated step outside the whole terrestrial ecosystem processes'? Do you actually mean that BVOC processes are not linked to photosynthesis? Another issue here is that the references the authors listed here were just based on one model. It is not a good idea to conclude this is a general problem for modelling BVOCs only based on one model.*
**Response:** Accepted. We reworded the sentence and added other references.
**Revisions:** (Page 2, Line 51) "The MEGAN algorithms have been incorporated into Community Land Model (CLM), the terrestrial component of the earth climate system model, as one step toward integrating biogeochemical processes in the model. In the coupling of MEGAN and CLM, all the physical and biological variables required by BVOC estimation are determined by comprehensive ecological and physiological processes parameterized in CLM at each time step (Levis et al., 2003; Oleson et al., 2010; Lawrence et al., 2011). Process-based models are typically coupled within dynamic vegetation models that have a mechanistic model for leaf photosynthesis at their core (Arneth et al., 2007; Pacifico et al., 2011; Yue and Unger, 2015). In general, these coupled models are employed to investigate the long-term interactions and feedbacks between terrestrial vegetation and climate change with spin-up and simulation time from tens to thousands of years.

Instead of coupling detailed algorithms within the land surface parameterizations, a simplified version of MEGAN algorithm, the parameterized canopy emission activity (PCEEA) algorithm, has been coupled with weather and climate forecasting models as an independent module to generate online biogenic emission inventory for atmospheric chemistry simulation (Guenther et al., 2006; Sakulyanontvittaya et al., 2008; Fu and Liao, 2012; Henrot et al., 2017). Instead of using a detailed canopy model to calculate leaf temperature and leaf-level photosynthetic photo flux

density (PPFD), the PCEEA algorithm parameterizes the modification of these plant physiological variables on emission rates based on ambient temperature and canopy above solar radiation. Although leaf temperature is strongly related to ambient temperature, it is also affected by other ecological conditions such as irradiation and evapotranspiration. Subin et al. (2011) indicated that the strong advection and boundary layer mixing during the day decoupled the air temperature from the vegetation temperature to a great extent, making daytime surface energy budget the primary controlling factors of vegetation temperature changes. Furthermore, due to the different morphological and physiological properties, relationships between air temperature and leaf temperature, and between canopy above PPFD and leaf-level PPFD, vary significantly among tree species. Since the PCEEA algorithm was based on standard MEGAN canopy model simulations for warm broadleaf forests, using the same equations for representations of other plant types leads to unpredictable uncertainties. Leaf temperature and PPFD averaged over the past 24 and 240 h are used in MEGAN algorithm to account for effects of medium-term weather history. However, the PCEEA algorithm obtains the past conditions from a prescribed climatological monthly mean dataset, which could be much different from the real meteorology (Zhao et al., 2016). Therefore, reasonable plant-specific physiological variables are needed to improve the BVOC estimation in weather models."

*L113-115, it is not clear for me why the authors decided to use MODIS LAI, instead of the modelled LAI and dynamic vegetation, and also why not consider CO2 impacts on emissions? Please clarify. MODIS LAI is 8 days intervals, how the author deal with the days between 2 LAI images. Another issue is that: did the authors consider quality flag for the LAI product?*

**Response:** The effects of variations in $CO_2$ concentration was neglected in this study because the simulation was performed for only one year and studies indicated that accounting for $CO_2$ inhibition has litter impact on predictions of present-day isoprene emission (Heald et al., 2009). The main purpose is to improve BVOC estimations in weather forecasting models by using plant-specific physiological parameters. Modeling dynamic vegetation was beyond the scope of this study and required more computing resources and more time for spin up. Compared with the prescribed monthly LAI in the default CLM4, MODIS data provide LAI data for a specific year with high spatiotemporal resolution. We used one LAI image for 8 days. Because the MODIS LAI Collection 6 product shows higher than 90 % main algorithm retrieval rates for most biome types (Yan et al., 2016) and relatively small influence of LAI as described in the Uncertainty section, the quality flag was not considered in our study.

*L121, ". . . and each run covered 31 days . . . " where these runs refer to? Please clarify.*

**Response:** Accepted. We reworded the sentences.

**Revisions:** (Page 6, Line 172) "The meteorological fields were initialized at the start of each model run, which covered one month in order to account for the effects of past canopy climate."

*L148, not clear for me what modification/process have been implemented.*

**Response:** Accepted. We reworded this section.

**Revisions:** (Page 4, Line 117) "The coupling of CLM4-MEGAN improves the BVOC estimations through reasonable driving factors and detailed sub-grid representation, as briefly introduced below. We refer the reader to the description of Oleson et al. (2010) for the details of computations.

1. leaf temperature

Variations in leaf temperature are influenced by net radiation absorbed/emitted by the vegetation and sensible and latent heat fluxes from vegetation. The two-stream approximation is applied to vegetation when calculating solar radiation reflected and absorbed by the canopy. Leaf temperatures are determined by the canopy energy balance equations. Due to the dependence of heat fluxes on vegetation temperature, the Newton-Raphson iteration is used to solve for folia temperature and the vegetation fluxes simultaneously.

2. sunlit and shaded fractions of canopy

The canopy in CLM4 is treated as sunlit and shaded leaves. The leaf fractions of different plant types are determined according to leaf and stem area index and the solar zenith angle at each time step. CLM4 assumed that sunlit leaves receive the absorbed direct beam radiation and the absorbed diffuse radiation apportioned by $f_{sun}$ (the sunlit fraction of the canopy), and that shaded leaves receive the absorbed diffuse radiation apportioned by $f_{sha}$ (the shaded fraction). This division into sunlit and shaded leaves is important in modeling canopy processes, since the sunlit leaves will receive a much higher light flux density than shaded leaves under sunny conditions (Dai et al., 2004).

3. the medium-term weather history

Current MEGAN algorisms use average leaf temperature, solar radiation and leaf fractions over the past 24 and 240 h to account for the influence of past canopy climate. CLM4 contains an accumulation module used to calculate the average of user-specified variables over user-defined time intervals. However, the accumulation of past time leaf temperature and PPFD are commented out in the default CLM4 code and fixed values are assigned to those coefficients based on conditions during previous days. After activating this module, we found a decrease in average temperature and PPFD with increasing simulation time. That was because these two variables were not being accumulated but still being averaged over the total running time. We corrected the accumulation code so that the average leaf temperature, PPFD and leaf fraction are calculated at each time step.

4. sub-gird heterogeneity

In CLM4, the surface heterogeneity is represented using a sub-grid tile approach in which grid cells are composed of multiple landunits (glacier, wetland, lake, urban and vegetated area), snow/soil columns and plant functional types (PFTs). Vegetated surfaces are comprised of up to 4 plant

functional types (PFTs). An explicit canopy layer represents the PFTs with specific leaf and stem optical properties, root distribution parameters, aerodynamic parameters and photosynthetic parameters. The detailed representations of sub-grid improve the accuracy of land surface parameterizations and lower the uncertainty from plant distribution in BVOC estimation (Zhao et al., 2016; Schultz et al., 2016)."

**References:**

Heald, C. L., Wilkinson, M. J., Monson, R. K., Alo, C. A., Wang, G., and Guenther, A.: Response of isoprene emission to ambient CO2 changes and implications for global budgets, Global Change Biology, 15, 1127-1140, 10.1111/j.1365-2486.2008.01802.x, 2009.

Yan, K., Park, T., Yan, G., Liu, Z., Yang, B., Chen, C., Nemani, R. R., Knyazikhin, Y., and Myneni, R. B.: Evaluation of MODIS LAI/FPAR Product Collection 6. Part 2: Validation and Intercomparison, Remote Sensing, 8, 10.3390/rs8060460, 2016.